# Efficient Contextual Bandits with Continuous Actions

**Maryam Majzoubi**[*]
New York University

**Chicheng Zhang**[†]
University of Arizona

**Rajan Chari**[‡]
Microsoft Research

**Akshay Krishnamurthy**[§]
Microsoft Research

**John Langford**[¶]
Microsoft Research

**Aleksandrs Slivkins**[‖]
Microsoft Research

## Abstract

We create a computationally tractable algorithm for contextual bandits with continuous actions having unknown structure. Our reduction-style algorithm composes with most supervised learning representations. We prove that it works in a general sense and verify the new functionality with large-scale experiments.

## 1 Introduction

In contextual bandit learning [6, 1, 39, 3], an agent repeatedly observes its environment, chooses an action, and receives a reward feedback, with the goal of optimizing cumulative reward. When the action space is discrete, there are many solutions to contextual bandit learning with successful deployments in personalized health, content recommendation, and elsewhere [e.g., 42, 54, 2, 44, 25, 43]. However, in many practical settings the action chosen is actually continuous. How then can we efficiently choose the best action given the context? This question is also extremely relevant to reinforcement learning more generally since contextual bandit learning is one-step reinforcement learning.

There are many concrete examples of reinforcement learning problems with continuous actions. In precision medicine [20, 31], doctors may prescribe to a patient a medication with a continuous value of dosage [32]. In data center optimization, the fan speeds and liquid coolant flow may be controllable continuous values [41]. In operating systems, when a computer makes a connection over the network, we may be able to adjust its packet send rate in response to the current network status [30]. All of these may be optimizable based on feedback and context.

A natural baseline approach here is to posit smoothness assumptions on the world, as in much prior work, e.g., [5, 34, 18, 50, 19]. This approach comes with practical drawbacks. Many applications do not exhibit any smoothness structure. When/if they do, the smoothness parameters (such as a Lipschitz constant) must be known in advance. Unfortunately, discovering the smoothness parameters is challenging, and requires knowing some other parameters and/or extensive exploration.

A recent approach to continuous actions [37] realizes similar performance guarantees without knowing the Lipschitz constant (let alone a more refined smoothness structure), while leveraging any preferred policy representation. Here, each action is "smoothed" to a distribution over an interval, and the benchmark one competes with is "smoothed" similarly. Unfortunately, their algorithm is computationally infeasible since it requires enumeration of all possible policy parameter settings.

---

[*]mm7918@nyu.edu
[†]chichengz@cs.arizona.edu
[‡]rajan.chari@microsoft.com
[§]akshaykr@microsoft.com
[¶]jcl@microsoft.com
[‖]slivkins@microsoft.com

In this paper, we realize benefits similar to this approach with a computationally practical algorithm, for contextual bandits with continuous action space $[0, 1]$. Our algorithms are *oracle-efficient* [39, 21, 3, 47, 53]: computationally efficient whenever we can solve certain supervised learning problems. Our main algorithm chooses actions by navigating a tree with supervised learners acting as routing functions in each node. Each leaf corresponds to an action, which is then "smoothed" to a distribution from which the final action is sampled. We use the reward feedback to update the supervised learners in the nodes to improve the "tree policy."

Our contributions can be summarized as follows:

- We propose `CATS`, a new algorithm for contextual bandits with continuous actions (Algorithm 1). It uses $\epsilon$-greedy exploration with tree policy classes (Definition 2) and is implemented in a fully online and oracle-efficient manner. We prove that `CATS` has prediction and update times scaling as log of the tree size, an exponential improvement over traditional approaches. Assuming realizability, `CATS` has a sublinear regret guarantee against the tree policy class (Theorem 6).

- We propose `CATS_Off`, an off-policy optimization version of `CATS` (Algorithm 3) that can utilize logged data to train and select tree policies of different complexities. We also establish statistical guarantees for this algorithm (Theorem 7).

- We implement our algorithms in Vowpal Wabbit (*vowpalwabbit.org*), and compare with baselines on real datasets. Experiments demonstrate the efficacy and efficiency of our approach (Section 5).

**Discussion.** The smoothing approach has several appealing properties. We look for a good interval of actions, which is possible even when the best single action is impossible to find. We need to guess a good *width*, but the algorithm adjusts to the best location for the interval. This is less guessing compared to uniform discretization (where the width and location are tied to some extent). While the bandwidth controls statistical performance, an algorithm is free to discretize actions for the sake of computational feasibility. An algorithm can improve accuracy by reusing datapoints for overlapping bands. Finally, the approach is principled, leading to specific, easily interpretable guarantees.

The tree-based classifier is a successful approach for supervised learning with a very large number of actions (which we need for computational feasibility). However, adapting it for smoothing runs into some challenges. First, a naive implementation leads to a prohibitively large per-round running time; we obtain an exponential improvement as detailed in Section 3.1. Second, existing statistical guarantees do not carry over to regret in bandits: they merely "transfer" errors from tree nodes to the root [10, 9], but the former errors could be huge. We posit a realizability assumption; even then, the analysis is non-trivial because the errors accumulate as we move down the tree.

Another key advantage of our approach is that it allows us to use off-policy model selection. For off-policy evaluation, we use smoothing to induce exploration distribution supported on the entire action space. Hence, we can discover when refinements in tree depth or smoothing parameters result in superior performance. Such model selection is not possible when using discretization approaches. When employed in an offline setup with data collected by a baseline *logging policy*, our experiments show that off-policy optimization can yield dramatic performance improvements.

**Related work.** Contextual bandits are quite well-understood for small, discrete action spaces, with rich theoretical results and successful deployments in practice. To handle large or infinite action spaces, most prior work either makes strong parametric assumptions such as linearity, or posits some continuity assumptions such as Lipschitzness. More background can be found in [17, 51, 40].

Bandits with Lipschitz assumptions were introduced in [5], and optimally solved in the worst case by [33]. [34, 35, 18, 50] achieve optimal data-dependent regret bounds, while several papers relax global smoothness assumptions with various local definitions [7, 34, 35, 18, 49, 45, 27]. This literature mainly focuses on the non-contextual version, except for [50, 36, 19, 56] (which only consider a fixed policy set $\Pi$). As argued in [37], the smoothing-based approach is productive in these settings, and extends far beyond, e.g., to instances when the global optimum is a discontinuity.

Most related to this paper is [37], which introduces the smoothness approach to contextual bandits and achieves data-dependent and bandwidth-adaptive regret bounds. Their approach extends to generic "smoothing distributions" (kernels), as well as to adversarial losses. However, their algorithms are inherently computationally inefficient, because they build on the techniques from [6, 21].

Our smoothing-based reward estimator was used in [37] for contextual bandits, as well as in [31, 20] in the observational setting. The works of [31, 20] learn policies that are linear functions of the context, and perform policy optimization via gradient descent on the IPS loss estimate.

## 2 Preliminaries

**Setting and key definitions.** We consider the stochastic (i.i.d.) contextual bandits (CB) setting. At each round $t$, the environment produces a (context, loss) pair $(x_t, \ell_t)$ from a distribution $\mathcal{D}$. Here, context $x_t$ is from the context space $\mathcal{X}$, and the loss function $\ell_t$ is a mapping from the action space $\mathcal{A} \triangleq [0, 1]$ to $[0, 1]$. Then, $x_t$ is revealed to the learner, based on which it chooses an action $a_t \in \mathcal{A}$ and observes loss $\ell_t(a_t)$. The learner's goal is to minimize its cumulative loss, $\sum_{t=1}^T \ell_t(a_t)$.

Define a *smoothing operator*: $\texttt{Smooth}_h : \mathcal{A} \to \Delta(\mathcal{A})$, that maps each action $a$ to a uniform distribution over the interval $\{a' \in \mathcal{A} : |a - a'| \leq h\} = [a - h, a + h] \cap [0, 1]$. As notation, let $\nu$ denote the Lebesgue measure, i.e. the uniform distribution over $[0, 1]$. Denote by $\texttt{Smooth}_h(a'|a)$ the probability density function w.r.t., $\nu$ for $\texttt{Smooth}_h(a)$ at action $a'$. We define $\texttt{Smooth}_0(a) \triangleq \delta_a$, where $\delta_a$ is the Dirac point mass at $a$. For a *policy* $\pi : \mathcal{X} \to \mathcal{A}$, we define $\pi_h(a'|x) \triangleq \texttt{Smooth}_h(a'|\pi(x))$ to be the probability density value for action $a'$ of the smoothed policy $\pi_h$ on context $x$.

Equivalently, we define $h$-smoothed loss $\ell_h(a) \triangleq \mathbb{E}_{a' \sim \texttt{Smooth}_h(\cdot|a)}[\ell(a')]$. For policy $\pi : \mathcal{X} \to \mathcal{A}$ we define the corresponding $h$*-smoothed expected loss* as $\lambda_h(\pi) \triangleq \mathbb{E}_{(x,\ell) \sim \mathcal{D}} \mathbb{E}_{a \sim \texttt{Smooth}_h(\pi(x))}[\ell(a)]$. This is equivalent to defining $\pi_h : x \mapsto \texttt{Smooth}_h(\pi(x))$, and evaluating $\pi_h$ on the original loss, i.e., $\lambda_0(\pi_h) = \lambda_h(\pi)$. The bandwidth $h$ governs an essential bias-variance trade-off in the continuous-action setting: with small $h$, the smoothed loss $\lambda_h(\pi)$ closely approximates the true expected loss function $\lambda_0(\pi)$, whereas the optimal performance guarantees scale inversely with $h$.

Over the $T$ rounds, the learner accumulates a history of interaction. After round $t$, this is $(x_s, a_s, p_s, \ell_s(a_s))_{s=1}^t$, where $x_s$ is the context, $a_s$ is the chosen action, $p_s = P_s(a_s \mid x_s)$ is the value of the density $P_s(\cdot \mid x_s)$ used at round $s$ at $a_s$, and $\ell_s(a_s)$ is the observed loss. From this history, we use an *inverse propensity score* (IPS) estimator [29] to compute an unbiased estimate of the smoothed loss $\lambda_h(\pi)$: $\hat{V}_t(\pi_h) = \frac{1}{t} \sum_{s=1}^t \frac{\pi_h(a_s|x_s)}{P_s(a_s|x_s)} \ell_s(a_s)$. A useful primitive in contextual bandits is to find a policy $\pi$ that minimizes $\hat{V}_t(\pi_h)$, which is a surrogate for $\lambda_0(\pi)$.

A natural approach for policy optimization is to reduce to cost-sensitive multiclass classification (CSMC). We choose a discretization parameter $K$, and instantiate a policy class $\Pi : \mathcal{X} \to \mathcal{A}_K$ where $\mathcal{A}_K \triangleq \{0, \frac{1}{K}, \frac{2}{K}, \dots, \frac{K-1}{K}\}$. Then, as $\pi_h(a_s \mid x_s) = \texttt{Smooth}_h(a_s \mid i/K)$, if $\pi(x_s) = i/K$, policy optimization can be naturally phrased as a CSMC problem. For each round, we create a cost-sensitive example $(x_s, \tilde{c}_s)$ where $\tilde{c}_s(i/K) = \frac{\ell_s(a_s)}{P_s(a_s|x_s)} \texttt{Smooth}_h(a_s|i/K)$, for all $i/K$ in $\mathcal{A}_K$. Then, optimizing $\hat{V}_t(\pi_h)$ is equivalent to computing $\text{argmin}_{\pi \in \Pi} \sum_{s=1}^t \tilde{c}_s(\pi(x_s))$. When working with $h$-smoothed losses, the error incurred by using the discretized action space $\mathcal{A}_K$ can be controlled, as we can show that $\ell_h(a)$ is $1/h$-Lipschitz [37].[1] So, this discretization strategy can compete with policies that are not restricted to $\mathcal{A}_K$, incurring an additional error of $\frac{1}{hK}$ per round.

**Tree policies.** One challenge with applying the CSMC approach is computational: for general classes $\Pi$, classical methods for CSMC (such as one-versus-all) have $\Omega(K)$ running time. This is particularly problematic since we want $K$ to be quite large in order to compete with policies that are not restricted to $\mathcal{A}_K$. To overcome this challenge, we consider a structured policy class induced by a binary tree $\mathcal{T}$, where each node $\mathtt{v}$ is associated with a binary classifier $f^{\mathtt{v}}$ from some base class $\mathcal{F}$.[2]

**Definition 1** (Tree policy). *Let $K = 2^D$ for some natural number $D$, and $\mathcal{F}$ be a class of binary classifiers from $\mathcal{X}$ to $\{\texttt{left}, \texttt{right}\}$. $\mathcal{T}$ is said to be a tree policy over action space $\mathcal{A}_K = \{i/K\}_{i=0}^{K-1}$ using $\mathcal{F}$, if: (1) $\mathcal{T}$ is a complete binary tree of depth $D$ with $K = 2^D$ leaves, where each leaf $\mathtt{v}$ has label $\text{label}(\mathtt{v}) = 0/K, \dots, K-1/K$ from left to right, respectively; (2) in each internal node $\mathtt{v}$ of $\mathcal{T}$, there is a classifier $f^{\mathtt{v}}$ in $\mathcal{F}$; (3) the prediction of $\mathcal{T}$ on an example $x$, $\mathcal{T}.\texttt{get\_action}(x)$, is defined*

*as follows. Starting from the root of $\mathcal{T}$, repeatedly route $x$ downward by entering the subtree that follows the prediction of the classifier in the tree nodes. When a leaf is reached, its label is returned (see Algorithm 4 in Appendix A for a formal description).*

In other words, a tree policy over action space $\mathcal{A}_K$ can be viewed as a decision tree of depth $D$, where its nodes form a hierarchical partition of the discretized action space $\mathcal{A}_K$. For each node in the tree, there is a subset of the context space that gets routed to it; therefore, given a tree policy $\mathcal{T}$ over $\mathcal{A}_K$, it also implicitly defines a hierarchical partition of the context space $\mathcal{X}$. The crucial difference between a tree policy and a decision tree in the usual sense, is that each leaf node corresponds to a distinct action. Our tree policy approach is also fundamentally different from the approach of [50, 56] in contextual bandits, in that their usages of trees are in performing regression of reward as a function of (context, action) pairs. Our policy classes of interest are tree policy classes:

**Definition 2.** *Let $\mathcal{F}_K$ denote the policy class of all tree policies over action space $\mathcal{A}_K$ using base class $\mathcal{F}$, that is, the set of tree policies $\mathcal{F}_K = \{\mathcal{T} : \mathcal{T}$ is a tree policy over action space $\mathcal{A}_K$ using $\mathcal{F}\}$. Furthermore, Let $\mathcal{F}_\infty$ denote the policy class of all tree policies of arbitrary depths using base class $\mathcal{F}$, formally, $\mathcal{F}_\infty = \bigcup_{K:K \in 2^{\mathbb{N}}} \mathcal{F}_K$.*

As a computational primitive, we assume that we can solve CSMC problems over the base class $\mathcal{F}$. Note that formally these are *binary* classification problems. The main advantage of using these structured policy classes is computational efficiency. As we demonstrate in the next section, we can use fast online CSMC algorithms to achieve a running time of $\mathcal{O}(\log K)$ per example. At the same time, due to the hierarchical structure, choosing an action using a policy in $\mathcal{F}_K$ also takes $\mathcal{O}(\log K)$ time. Both of these are exponentially faster than the $\mathcal{O}(K)$ running time that typically arises from flat representations. Finally, given a tree policy, we define the tree policy rooted at one of its nodes:

**Definition 3.** *Let $\mathtt{v}$ be an internal node in $\mathcal{T}$. We define $\mathcal{T}^{\mathtt{v}}$ as the tree-based policy with root at $\mathtt{v}$. We will abbreviate $\mathcal{T}^{\mathtt{v}}.\mathtt{get\_action}(x)$ as $\mathcal{T}^{\mathtt{v}}(x)$ or $\mathtt{v.get\_action}(x)$.*

**The performance benchmark.** We define the performance benchmark: $\mathrm{Reg}(T, \mathcal{F}_\infty, h) \triangleq \mathbb{E}\left[\sum_{t=1}^{T} \ell_t(a_t)\right] - T\inf_{\pi \in \mathcal{F}_\infty} \lambda_h(\pi)$. In words, we are comparing the cumulative expected loss of our algorithm, with the *h-smoothed* cumulative expected loss of the best tree policy of arbitrary depth. We call this the *h-smoothed regret* w.r.t. $\mathcal{F}_\infty$. Although the focus of this paper is on contextual bandit algorithms with computational efficiency guarantees, in Appendix D, we also present several extensions of our results to general policy classes.

**Miscellaneous notation.** Given a set of CSMC examples $S = \{(x, c)\}$ of size $n$, and a function $f$, we use $\mathbb{E}_S[f(x, c)] \triangleq \frac{1}{n} \sum_{(x,c) \in S} f(x, c)$ to denote empirical expectation of $f$ over $S$. Given a function $f$ with domain $\mathcal{Z}$, define its range to be the set of values it can take, i.e. $\mathrm{range}(f) \triangleq \{f(z) : z \in \mathcal{Z}\}$. Specifically, given a tree $\mathcal{T}$ over action space $\mathcal{A}_K$ and a node $\mathtt{v}$ in $\mathcal{T}$, $\mathrm{range}(\mathcal{T}^{\mathtt{v}})$ denotes the actions reachable by $\mathcal{T}$, i.e. the action labels of the leaves that are descendants of $\mathtt{v}$. Given a natural number $n$, we denote by $[n] \triangleq \{1, \ldots, n\}$.

## 3 Algorithm

We describe our main algorithm CATS for learning with continuous actions using tree policies in Algorithm 1 and an off-policy version in Algorithm 3 for unknown $h$. In Appendix C, we also present a variant of CATS that works online for unknown $h$.

### 3.1 Smoothed $\epsilon$-greedy algorithm with trees

We present Algorithm 1 in this section. It consists of two main components: first, a smoothed $\epsilon$-greedy exploration strategy (lines 4 to 6); second, a tree training procedure Train_tree called at line 7, namely Algorithm 2. We discuss each component in detail next.

**$\epsilon$-greedy exploration with smoothing.** At time step $t$, the algorithm uses the policy $\pi_t$ learned from data collected in previous time steps to perform action selection. Specifically, with probability $\epsilon$, it chooses an action uniformly at random from $\mathcal{A}$; otherwise, it chooses an action based on the prediction of $\pi_{t,h}$, the $h$-smoothing of policy $\pi_t$. As we will see, $\pi_{t,h}$ has expected loss competitive

---

**Algorithm 1** CATS: continuous action tree with smoothing

---

**Input:** Exploration and smoothing parameters $(\epsilon, h)$, discretization scale $K = 2^D$, base class $\mathcal{F}$
1: Initialize dataset $S_0 \leftarrow \emptyset$, and tree policy $\mathcal{T}$ with classifiers $f^{\mathsf{v}} \in \mathcal{F}$ at every internal node $\mathsf{v}$.
2: **for** $t = 1, 2, \ldots, T$ **do**
3:   Let $\pi_t$ be the tree policy $\mathcal{T}$ with $\{f^{\mathsf{v}}\}$ as classifiers.
4:   Define policy $P_t(a \mid x) := (1 - \epsilon)\pi_{t,h}(a|x) + \epsilon$.
5:   Observe context $x_t$, select action $a_t \sim P_t(\cdot \mid x_t)$, observe loss $\ell_t(a_t)$.
6:   Let $\tilde{c}_t(i/K) \leftarrow \frac{\texttt{Smooth}_h(a_t|i/K)}{P_t(a_t|x_t)} \ell_t(a_t)$ for all $i$
7:   $\mathcal{T} \leftarrow \texttt{Train\_tree}(K, \mathcal{F}, \{(x_s, \tilde{c}_s)\}_{s=1}^t)$.

---

**Algorithm 2** Tree training: `Train_tree`

---

**Input:** $K = 2^D$, $\mathcal{F}$, data $\{(x_s, c_s)\}_{s=1}^n$ with $c_s \in \mathbb{R}^K$
1: For level $d = 0, \ldots, D - 1$: $B_d \leftarrow \{(D - d - 1)n' + 1, \ldots, (D - d)n'\}$, where $n' = \lfloor n/D \rfloor$.
2: **for** level $d$ from $D - 1$ down to $0$ **do**
3:   **for** nodes $\mathsf{v}$ at level $d$ **do**
4:     For each $(x_s, c_s)$ define binary cost $c_s^{\mathsf{v}}$ with
       $c_s^{\mathsf{v}}(\texttt{left}) = c_s(\mathsf{v}.\texttt{left.get\_action}(x_s)), c_s^{\mathsf{v}}(\texttt{right}) = c_s(\mathsf{v}.\texttt{right.get\_action}(x_s))$.
5:     Train classifier at node $\mathsf{v}$: $f^{\mathsf{v}} \leftarrow \arg\min_{f \in \mathcal{F}} \mathbb{E}_{S^{\mathsf{v}}}[c^{\mathsf{v}}(f(x))]$, where $S^{\mathsf{v}} = \{(x_s, c_s^{\mathsf{v}}) : s \in B_l, c_s^{\mathsf{v}}(\texttt{left}) \neq c_s^{\mathsf{v}}(\texttt{right})\}$:
6: **return** tree $\mathcal{T}$ with $\{f^{\mathsf{v}}\}$ as node classifiers.

---

with any smoothed policy $\pi_h$ with $\pi$ in $\mathcal{F}_K$ (and is therefore competitive with $\mathcal{F}_\infty$). This component is similar to the $\epsilon$-greedy algorithm for discrete action contextual bandits [e.g. 39]; here $\epsilon$ is a parameter that trades off between exploration and exploitation, where a larger $\epsilon$ yields better quality data for learning, and a smaller $\epsilon$ implies actions with better instantaneous losses are taken.

**Tree training.** Given the interaction log collected up to time $t$, $\{(x_s, a_s, P_s(a_s \mid x_s), \ell_t(a_s))\}_{s=1}^t$, Algorithm 1 incorporates it to produce a policy $\pi_{t+1}$ for time $t + 1$. Specifically, $\pi_{t+1}$ is a tree policy $\mathcal{T}$ in $\mathcal{F}_K$ that approximately minimizes $\hat{V}_t(\mathcal{T}_h')$ over all policies $\mathcal{T}'$ in $\mathcal{F}_\infty$. To this end, we use `Train_tree` (Algorithm 2) over the set of cost-sensitive examples $(x_s, \tilde{c}_s)_{s=1}^t$ constructed by IPS. For technical reasons[3], `Train_tree` differs from the filter tree algorithm [10] in that it partitions the dataset into $D = \log K$ subsets, with their indices $B_0, \ldots, B_{D-1}$ being disjoint subsets in $[n]$. For every $i$, the examples with indices in $B_i$ are dedicated to training classifiers in tree nodes at level $i$. `Train_tree` trains the classifiers in the tree nodes in a bottom-up fashion. At the bottom layer, each node $\mathsf{v}$ with two leaves as children seeks a classifier $f_{\mathsf{v}}$ in $\mathcal{F}$ that directly classifies the context $x$ to the action in $\text{range}(\mathcal{T}^{\mathsf{v}})$ with smaller expected cost $\mathbb{E}[\ell(a) \mid x]$. For this, it invokes CSMC learning with class $\mathcal{F}$ where costs are the IPS costs for the two children $\mathsf{v}.\texttt{left}$ and $\mathsf{v}.\texttt{right}$. At other internal nodes $\mathsf{v}$, given that all the downstream classifiers in subtrees rooted at $\mathsf{v}.\texttt{left}$ and $\mathsf{v}.\texttt{right}$ have been trained, it aims to find a classifier $f_{\mathsf{v}}$ in $\mathcal{F}$ such that $f_{\mathsf{v}}$, in conjunction with other classifiers in $\mathcal{T}^{\mathsf{v}}$, routes context $x$ to the action in $\text{range}(\mathcal{T}^{\mathsf{v}})$ with the smallest expected cost.

**Computational complexity.** CATS can be implemented in a fully online and oracle-efficient fashion, using online CSMC learners. Specifically, line 5 in `Train_tree` can be implemented by maintaining a stateful online learner for each tree node $\mathsf{v}$, which at time $t$ maintains $f_t^{\mathsf{v}}$, an approximation of $\arg\min_{f \in \mathcal{F}} \sum_{s=1}^{t-1} [c_s^{\mathsf{v}}(f(x_s))]$. Then, upon seeing a binary CSMC example $(x_t, c_t^{\mathsf{v}})$, the learner employs incremental update rules such as stochastic gradient descent to update its internal state to $f_{t+1}^{\mathsf{v}}$, an approximate solution to the next CSMC problem.

We now look at the per-example computational cost of CATS using the above online implementation of CSMC oracle. Naively, in line 5 of `Train_tree`, if we instead define $S^{\mathsf{v}} = \{(x_s, c_s^{\mathsf{v}}) : s \in B_i\}$ for every node $\mathsf{v}$, i.e. we do not filter out examples with identical costs for left and right sides at

**Algorithm 3** `CATS_Off`

---

**Input:** logged data $\{(x_t, a_t, P_t(a_t \mid x_t), \ell_t(a_t))\}_{t=1}^{T}$, minimum density of action distribution $p_{\min}$,
set of (bandwidth, discretization level) combinations $\mathcal{J} \subset [0,1] \times 2^{\mathbb{N}}$, base class $\mathcal{F}$.

1: **for** (bandwidth, discretization level) $(h, K)$ in $\mathcal{J}$ **do**

2:     For every $t$ in $[T]$, let $\tilde{c}_t^h(i/K) \leftarrow \frac{\texttt{Smooth}_h(a_t|i/K)}{P_t(a_t|x_t)} \ell_t(a_t)$ for all $i \in \{0, \dots, K-1\}$.

3:     **for** $t = 1, 2, \dots, T$ **do**

4:         $\mathcal{T}_t^{h,K} \leftarrow \texttt{Train\_tree}(K, \mathcal{F}, \{(x_s, \tilde{c}_s^h)\}_{s=1}^{t-1})$.

5: Let $(\hat{h}, \hat{K}) \leftarrow \operatorname{argmin}_{(h,K) \in \mathcal{J}} \left( \frac{1}{T} \sum_{t=1}^{T} \tilde{c}_t^h(\mathcal{T}_t^{h,K}(x_t)) + \text{Pen}(h, K) \right)$, where $\text{Pen}(h, K) =$

    $\sqrt{\frac{1}{T} \sum_{t=1}^{T} \tilde{c}_t^h(\mathcal{T}_t^{h,K}(x_t)) \cdot \frac{64 \ln \frac{4T|\mathcal{J}|}{\delta}}{T p_{\min} h}} + \frac{64 \ln \frac{4T|\mathcal{J}|}{\delta}}{T p_{\min} h}$.

6: **return** $\hat{\pi}$ drawn uniformly at random over set $\left\{ \mathcal{T}_{t,\hat{h}}^{\hat{h},\hat{K}} \right\}_{t=1}^{T}$.

---

node v, the time for processing each example would be $\mathcal{O}(K)$, since it contributes a binary CSMC example to $S^{\texttt{v}}$ for every node v.

Our first observation is that, if at time $t$, $c_t^{\texttt{v}}(\texttt{left}) = c_t^{\texttt{v}}(\texttt{right})$, the online CSMC learner can skip processing example $(x_t, c_t^{\texttt{v}})$, as is done in line 5 of `Train_tree`. This is because adding this example does not change the cost-sensitive ERM from round $t$ to round $t+1$. However, the algorithm still must decide whether this happens for each node v, which still requires $\mathcal{O}(K)$ time naively.

Our second observation is that, by carefully utilizing the piecewise constant nature of the IPS cost vector $\tilde{c}_t$, we can find the nodes that need to be updated and compute the costs of their left and right children, both in $\mathcal{O}(\log K)$ time per example. Specifically, as $\tilde{c}_t$ is piecewise constant with two discontinuities, only two root-to-leaf paths contain nodes that have children with differing costs and must be updated (see Appendix G, specifically Lemma 17 and its proof for more explanations). Exploiting these observations, we implement CATS to have $\mathcal{O}(\log K)$ update time, an exponential improvement over naive implementations. This is summarized in the next theorem.

**Theorem 4.** CATS *with an online learner at each node requires* $\mathcal{O}(\log K)$ *computation per example.*

We elaborate on our online implementation and present the proof of the theorem in Appendix G. Our theorem generalizes the computational time analysis of the offset tree algorithm for discrete-action contextual bandits [9], in that we allow input IPS CSMC examples to have multiple nonzero entries.

### 3.2 Off-policy optimization

As discussed above, one major advantage of the smoothing approach to contextual bandits with continuous actions is that policy optimization can be easily reduced to a CSMC learning problem via counterfactual techniques. This allows off-policy optimization, in the sense that the logged data can be collected using one policy that takes action in $\mathcal{A}$, while we can optimize over (smoothed) policy classes that take actions in $\mathcal{A}_K$. In the special setting that we learn from a tree policy class $\mathcal{F}_K$, the `CATS_Off` algorithm (Algorithm 3) can be used.

The algorithm receives an interaction log $\{(x_t, a_t, P_t(a_t \mid x_t), \ell_t(a_t))\}_{t=1}^{T}$, collected by another algorithm such that $P_t(a \mid x_t) \geq p_{\min}$ for all $a \in \mathcal{A}$, a collection of (bandwidth, disretization levels) $\mathcal{J}$, and a base policy class $\mathcal{F}$ as input. It consists of two stages: tree training and policy selection. In the tree training stage (lines 1 to 4), for each $((h, K), t)$ combination in $\mathcal{J} \times [T]$, the algorithm again calls `Train_tree` over cost-sensitive examples $\{(x_s, \tilde{c}_s^h)\}_{s=1}^{t-1}$ induced by the interaction log and the bandwidth $h$. As a result, we obtain a set of tree policies $\left\{ \mathcal{T}_t^{h,K} : t \in [T], (h, K) \in \mathcal{J} \right\}$ In the policy selection stage (line 5), we choose a pair $(\hat{h}, \hat{K})$ from the set $\mathcal{J}$ using structural risk minimization [55], by trading off $\frac{1}{T} \sum_{t=1}^{T} \tilde{c}_t^h(\mathcal{T}_t^{h,K}(x_t))$, the progressive validation loss estimate of smoothed policies $\left\{ \mathcal{T}_{t,h}^{h,K} : t \in [T] \right\}$ on logged data [15] and its deviation bound $\text{Pen}(h, K)$ that depends on $h$ and $K$. A similar procedure has been proposed in the discrete-action contextual bandit

learning setting [52]. As we see from Theorem 7 below, the obtained tree policy $\mathcal{T}$ has expected loss competitive with all policies in the set $\cup_{(h,K)\in\mathcal{J}}\{\pi_h : \pi \in \mathcal{F}_K\}$.

## 4 Performance guarantees

In this section, we show that CATS and CATS_Off achieve sublinear regret or excess loss guarantees under a realizability assumption over the (context, loss) distribution $\mathcal{D}$. We defer the formal statements of our theorems and their proofs to Appendix B.

As learning decision trees is computationally hard in general [28], many existing positive results pose strong assumptions on the learning model, such as uniform or product unlabeled distribution [13, 16], separability [23, 48, 14] or allowing membership queries [38, 26]. Our tree policy training guarantee under the following realizability assumption is complementary to these works:

**Definition 5.** *A hypothesis class $\mathcal{F}$ and data distribution $\mathcal{D}$ is said to be $(h, K)$-realizable, if there exists a tree policy $\mathcal{T}$ in $\mathcal{F}_K$ such that the following holds: for every internal node $\mathtt{v}$ in $\mathcal{T}$, there exists a classifier $f^{\mathtt{v},\star}$ in $\mathcal{F}$, such that*

$$f^{\mathtt{v},\star}(x) = \mathtt{left} \Rightarrow \min_{a \in \mathrm{range}(\mathcal{T}^{\mathtt{l}})} \mathbb{E}[\ell_h(a) \mid x] \leq \min_{a \in \mathrm{range}(\mathcal{T}^{\mathtt{r}})} \mathbb{E}[\ell_h(a) \mid x],$$

$$f^{\mathtt{v},\star}(x) = \mathtt{right} \Rightarrow \min_{a \in \mathrm{range}(\mathcal{T}^{\mathtt{r}})} \mathbb{E}[\ell_h(a) \mid x] \leq \min_{a \in \mathrm{range}(\mathcal{T}^{\mathtt{l}})} \mathbb{E}[\ell_h(a) \mid x],$$

*where $\mathtt{l} = \mathtt{v.left}$ and $\mathtt{r} = \mathtt{v.right}$ are $\mathtt{v}$'s two children; recall that $\ell_h(a) \triangleq \mathbb{E}_{a' \sim \mathtt{Smooth}_h(\cdot|a)}\ell(a')$.*

Intuitively, the above realizability assumption states that our base class $\mathcal{F}$ is expressive enough, such that for every discretization parameter $K$ in $2^{\mathbb{N}}$, there exists a set of $(K-1)$ classifiers $\{f^{\mathtt{v},\star}\}_{\mathtt{v}\in\mathcal{T}} \subset \mathcal{F}$ occupying the internal nodes of a tree $\mathcal{T}$ of $K$ leaves, and $\mathcal{T}$ routes any context $x$ to its Bayes optimal discretized action in $\mathcal{A}_K$, formally $\mathrm{argmin}_{a\in\mathcal{A}_K} \mathbb{E}[\ell_h(a) \mid x]$. As $\ell_h(\cdot)$ is $\frac{1}{h}$-Lipschitz, $\min_{a\in\mathcal{A}_K} \mathbb{E}[\ell_h(a) \mid x] - \min_{a\in\mathcal{A}} \mathbb{E}[\ell_h(a) \mid x] \leq \frac{1}{hK}$. This implies that, if $K$ is large enough, the Bayes optimal policy $\pi^{\star}(x) = \mathrm{argmin}_{a\in\mathcal{A}} \mathbb{E}[\ell_h(a) \mid x]$ can be well-approximated by a tree policy in $\mathcal{F}_K$ with little excess loss. Under the above realizability assumption, we now present a theorem that characterizes the regret guarantee when Algorithm 1 uses policy class $\mathcal{F}_K$.

**Theorem 6** (Informal). *Given $h$, suppose $(\mathcal{F},\mathcal{D})$ is $(h, K)$-realizable for any $K \in 2^{\mathbb{N}}$. Then with appropriate settings of greedy parameter $\epsilon$ and discretization scale $K$, with high probability, Algorithm 1 run with inputs $\epsilon, h, K, \mathcal{F}$ has regret bounded as: $\mathrm{Reg}(T, \mathcal{F}_\infty, h) \leq \mathcal{O}\Big( \big( T^4 \ln|\mathcal{F}|/h^3 \big)^{1/5} \Big)$.*

We remark that we actually obtain a stronger result: with appropriate tuning of $\epsilon$, the $h$-smoothed regret of CATS against $\mathcal{F}_K$ is $\mathcal{O}\bigg( \Big( K^2 T^2 \ln\frac{|\mathcal{F}|}{\delta}/h \Big)^{1/3} \bigg)$, which is similar to the $\mathcal{O}\big( T^{2/3}|\mathcal{A}|^{1/3} \big)$ regret for $\epsilon$-greedy in the discrete actions setting, with $1/h$ serving as the "effective number of actions." We also note that if we used exact ERM over $\mathcal{F}_K$ instead of the computationally efficient Train_tree procedure, the dependence on $K$ would improve from $K^{2/3}$ to $K^{1/3}$. This $K^{2/3}$ dependence is due to compounding errors accumulating in each node, and we conjecture that it is the price we have to pay for using the computationally-efficient Train_tree for approximate ERM.

The aforementioned $h$-smoothed regret bound against $\mathcal{F}_K$ reflects a natural bias-variance tradeoff in the choice of $h$ and $K$: for a smaller value of $h$, the $h$-smoothed loss more closely approximates the true loss, while achieving a low $h$-smoothed regret bound is harder. A similar reasoning applies to $K$: For larger $K$, $\mathcal{F}_K$ more closely approximates $\mathcal{F}_\infty$, while the regret of CATS against $\mathcal{F}_K$ can be higher.

We now present learning guarantees of CATS_Off under the same realizability assumption.

**Theorem 7** (Informal). *Suppose $(\mathcal{F},\mathcal{D})$ is $(h, K)$-realizable for all $(h, K) \in \mathcal{J}$. In addition, the logged data $S = \{(x_t, a_t, P_t(a_t \mid x_t), \ell_t(a_t))\}_{t=1}^{T}$ has a sufficient amount of exploration: $P_t(a \mid x_t) \geq p_{\min}$. Then, with high probability, Algorithm 3 run with inputs $S, p_{\min}, \mathcal{J}, \mathcal{F}$ outputs a policy $\hat{\pi}$ such that: $\lambda_0(\hat{\pi}) \leq \min_{(h,K)\in\mathcal{J},\pi\in\mathcal{F}_K} \bigg( \lambda_h(\pi) + \mathcal{O}\Big( K\sqrt{\ln\frac{|\mathcal{F}||\mathcal{J}|}{\delta}/(p_{\min}hT)} \Big) \bigg).$*

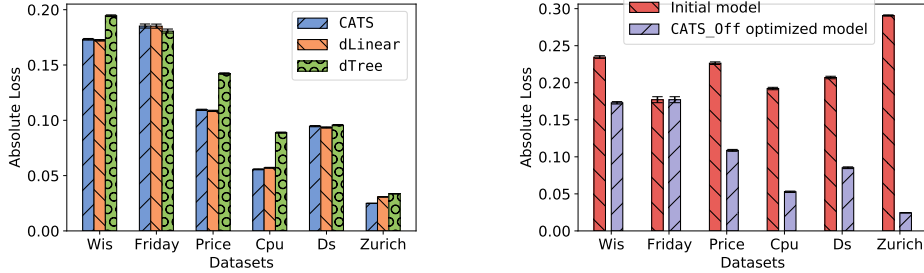

Figure 1: (**left**) Best progressive validation losses obtained by parameter search for different online learning algorithms on six regression datasets. (**right**) Test-set absolute losses for initial online-trained model using `CATS` with an initial set of discretization and smoothing parameter $(K_{\texttt{init}}, h_{\texttt{init}}) = (4, 1/4)$, and off-policy optimized models output by `CATS_Off`. All confidence intervals are calculated with a single run using the Clopper-Pearson interval with 95% confidence level (note that they are very small for most of the datasets).

The above theorem shows the adaptivity of `CATS_Off`: so long as the logged data is generated by an sufficiently explorative logging policy, its learned policy is competitive with any policy in the set $\cup_{(h,K) \in \mathcal{J}} \{\pi_h : \pi \in \mathcal{F}_K\}$, under realizability assumptions.

## 5    Experiments

Following the contextual bandit learning evaluation protocol of [12], we evaluate our approach on six large-scale regression datasets, where regression predictions are treated as continuous actions in $\mathcal{A} = [0, 1]$. To simulate contextual bandit learning, we first perform scaling and offsetting to ensure $y_t$'s are also in $[0, 1]$. Every regression example $(x_t, y_t)$ is converted to $(x_t, \ell_t)$, where $\ell_t(a) = |a - y_t|$ is the absolute loss induced by $y_t$. When action $a_t$ is taken, the algorithm receives bandit feedback $\ell_t(a_t)$, as opposed to the usual label $y_t$.

Of the six datasets, five are selected from OpenML with the criterion of having millions of samples with unique regression values (See Appendix F for more details). We also include a synthetic dataset ds, created by the linear regression model with additive Gaussian noise.

**Online contextual bandit learning using `CATS`.** We compare `CATS` with two baselines that perform $\epsilon$-greedy contextual bandit learning [39] over the discretized action space $\mathcal{A}_K$. The first baseline, `dLinear`, reduces policy training to cost-sensitive one-versus-all multiclass classification [11] which takes $\mathcal{O}(K)$ time per example. The second baseline, `dTree`, uses the filter tree algorithm [10] as a cost-sensitive multiclass learner for policy training, which takes $\mathcal{O}(\log K)$ time per example, but does not perform information sharing among actions through smoothing. We run `CATS` with $(h, K)$ combinations in the following set:

$$\mathcal{J} = \left\{ (h, K) : h \in \left\{ 2^{-13}, \dots, 2^{-1} \right\}, K \in \left\{ 2^2, \cdots, 2^{13} \right\}, hK \in \left\{ 2^0, \dots, 2^{11} \right\} \right\}. \tag{1}$$

We also run `dLinear` and `dTree` with values of $K$ in $\left\{ 2^1, 2^2, \cdots, 2^{13} \right\}$. All algorithms use $\epsilon = 0.05$; see Appendix F for additional experimental details.

In the left panel of Figure 1 we compare `CATS` with `dLinear` and `dTree`. Using progressive validation [15] for online evaluation, our algorithm (with optimally-tuned discretization and bandwidth) achieves performance similar to `dLinear`, and is better than `dTree` for most of the datasets.

As discussed in Section 3, the time cost of our implementation of `CATS` is $\mathcal{O}(\log(K))$ per example. Figure 2 demonstrates that the training time of `CATS` is constant w.r.t. bandwidth $h$, and grows logarithmically w.r.t. the discretization $K$. This shows that `CATS` has the same computational complexity as `dTree`. In contrast, `dLinear` has $\mathcal{O}(K)$ time complexity per example. The time improvement of `CATS` compared with `dLinear` becomes more significant when $K$ becomes larger. In summary, `CATS` outperforms `dTree` statistically, and has much better scalability than `dLinear`.

**Off-policy optimization using `CATS_Off`.** A major advantage of the `CATS` approach over naïve discretization methods is that the interaction log collected by our algorithm with one setting of $(h, K)$ can be used to optimize policies with alternate settings of $(h, K)$. To validate this, we first

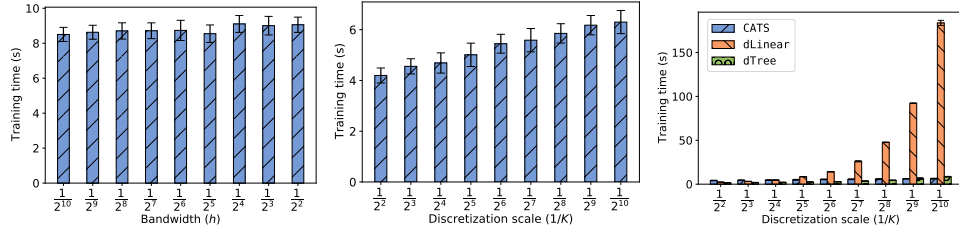

Figure 2: Online learning time costs of `CATS` (blue bar) w.r.t: (**left**) bandwidth ($h$) with a fixed discretization scale $K = 2^{13}$; (**middle**) discretization scale ($1/K$) with a fixed $h = 1/4$; (**right**) discretization scale ($1/K$) with a fixed $h = 1/4$, compared against `dLinear` (orange bar) and `dTree` (green bar), on the `ds` dataset. Similar figures for the rest of the datasets can be found in the Appendix H.

create an 80-20% split or training and test sets. With the training set, we first collect interaction log tuples of $(x_t, a_t, P_t(a_t \mid x_t), \ell_t(a_t))$ using `CATS` with initial discretization and smoothing parameter $(K_{\text{init}}, h_{\text{init}}) = (4, \frac{1}{4})$, and greedy parameter $\epsilon = 0.05$. We then run `CATS_Off` over the logged data using $\mathcal{J}$, defined in (1), as the set of parameters. Since standard generalization error bounds are loose in practice, we replaced $64 \ln \frac{2T|\mathcal{J}|}{\delta}$ in the penalty term in line 5 with constant 1. Note that this constant term as well as the learning rate and the greedy parameter are fixed for all of the datasets in our experiments.

The right panel in Figure 1 shows the test losses of the models obtained by `CATS` after making a pass over the training data, and the test losses of the optimized models obtained through `CATS_Off` by optimizing counterfactual estimates offline. It can be seen that offline policy training produces tree policies that have dramatically smaller test losses than the original policies.

## 6 Conclusion

Contextual bandit learning with continuous actions with unknown structure is quite tractable via the `CATS` algorithm, as we have shown theoretically and empirically. This broadly enables deployment of contextual bandit approaches across a wide range of new applications.

## Broader Impact

Our study of efficient contextual bandits with continuous actions can be applied to a wide range of applications, such as precision medicine, personalized recommendations, data center optimization, operating systems, networking, etc. Many of these applications have potential for significant positive impact to society, but these methods can also cause unintend harms, for example by creating filter bubble effects when deployed in recommendation engines. More generally our research belongs to the general paradigm of interactive machine learning, which must always be used with care due to the presence of feedback loops. We are certainly mindful of these issues, and encourage practitioners to consider these consequences when deploying interactive learning systems.

## Acknowledgments and Disclosure of Funding

We thank the anonymous reviewers for their helpful feedback. Much of this work was done while Maryam Majzoubi and Chicheng Zhang were visiting Microsoft Research NYC. This work was supported by Microsoft.

## Footnotes

[1] Although we use the $1/h$-Lipschitz property of $h$-smoothed losses here, in general, $h$-smoothed losses have more structure than $1/h$-Lipschitz losses, which admit better regret guarantees in general.

[2] We assume that $\mathcal{F}$ is finite for simplicity. This can be extended with empirical process arguments.

[3]We need to partition the input CSMC dataset in a delicate manner to ensure `Train_tree`'s theoretical guarantees; see Lemma 9 and its proof in Appendix B for more details. In our implementation we ignore such subtlety; see Algorithm 8 in Appendix G.

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
