[Supplementary Material · 6514-supplement.pdf]

---

**Algorithm 4** Execution of tree policy $\mathcal{T}$: $\mathcal{T}$.get_action

---

**Input:** Tree policy $\mathcal{T}$ using classifiers $\{f^{\mathtt{v}}\} \subset (\mathcal{F} \to \{\mathtt{left}, \mathtt{right}\})$, context $x$.
  Let $\mathtt{v} \leftarrow \mathcal{T}.\mathtt{root}$.
  **while** $\mathtt{v}$ is an internal node of $\mathcal{T}$ **do**
    $\mathtt{v} \leftarrow \mathtt{v}.f^{\mathtt{v}}(x)$
  **return** $a \leftarrow \mathrm{label}(\mathtt{v})$, the action label of $\mathtt{v}$.

---

## A  Additional Notation

Throughout the appendices, we will use all notation from Section 2, without further recap, as well as some additional notation presented below. For a policy $\pi$, define $V(\pi) = \lambda_0(\pi) = \mathbb{E}_{(x,\ell)\sim\mathcal{D}}\mathbb{E}_{a\sim\pi(\cdot|x)}[\ell(a)]$ to be its expected loss. We will use the notations $V(\pi)$ and $\lambda_0(\pi)$ interchangably throughout the appendix.

For a subset of indices $B \subset [n]$ and a policy $\pi$, denote by $\hat{V}_B(\pi_h) = \frac{1}{|B|} \sum_{s\in B} \frac{\pi_h(a_s|x_s)}{P_s(a_s|x_s)} \ell_s(a_s)$.

For a general policy class $\Pi \subset (\mathcal{X} \to \mathcal{A})$, we define the $h$-smoothed regret of an algorithm against $\Pi$ for a time horizon of $T$ as:

$$\mathrm{Reg}(T, \Pi, h) \triangleq \sum_{t=1}^{T} \mathbb{E}\left[\ell_t(a_t)\right] - T \inf_{\pi\in\Pi} \lambda_h(\pi) = \sum_{t=1}^{T} \mathbb{E}\left[\ell_t(a_t)\right] - T \inf_{\pi\in\Pi} V(\pi_h).$$

We will be using the following property of logged data, which has the essential independence structure to guarantee the quality of the model trained with $\mathtt{Train\_tree}$ on its induced CSMC examples using IPS.

**Definition 8** (Well-formed logged data). *The logged data $\{(x_s, a_s, P_s(a_s \mid x_s), \ell_s(a_s))\}_{s=1}^{n}$ is said to be $p_{\min}$-well-formed, if it is generated by the following process: $(x_s, \ell_s)_{s=1}^{n}$ are drawn iid from $\mathcal{D}$, action distribution $P_s(\cdot \mid \cdot)$ depends only on $(x_{s'}, a_{s'}, \ell_{s'})_{s'=1}^{s-1}$, $P_s(a \mid x) \geq p_{\min}$ for all $a \in \mathcal{A}$, $x \in \mathcal{X}$, and $s \in [n]$.*

A formal description of the execution of tree policies, i.e. $\mathcal{T}$.get_action$(x)$, is given in Algorithm 4.

## B  Proofs of Theorems 6 and 7

In this section, we first prove a key lemma, namely Lemma 9, and use it to show Theorems 6 and 7 in the main text respectively.

### B.1  Off-policy optimization guarantees on trees with well-formed logged data

Recall that $\mathcal{F}$ is a class of binary classifiers, and $\mathcal{D}$ is a distribution over (context, loss) pairs. In words, this lemma states that, under realizability and the well-formedness property of the logged data, training using $\mathtt{Train\_tree}$ based on its induced IPS CSMC examples yields a tree policy that has a $h$-smoothed loss competitive with any tree policy in tree class $\mathcal{F}_K$.

**Lemma 9** (Off-policy optimization with tree classes under realizability). *Suppose:*

1. $(\mathcal{F}, \mathcal{D})$ *is $(h, K)$-realizable for $h > 0$, $K = 2^D$ for some $D$ in $\mathbb{N}$.*

2. *The logged data $\{(x_s, a_s, P_s(a_s \mid x_s), \ell_s(a_s))\}_{s=1}^{n}$ is $p_{\min}$-well-formed.*

*In addition, Algorithm 2 is run with dataset $S = \{(x_s, \tilde{c}_s)\}_{s=1}^{n}$ (a set of CSMC examples induced by the logged data using IPS; see Section 2 for the definition of $\tilde{c}_s$), bandwidth $h$, discretization level $K$, base class $\mathcal{F}$. Then, with probability $1 - \delta$, the policy $\mathcal{T}$ returned is such that:*

$$V(\mathcal{T}_h) \leq \min_{\mathcal{T}'\in\mathcal{F}_K} V(\mathcal{T}'_h) + 20\sqrt{\frac{K^2 \log K}{np_{\min}h} \cdot \left(\ln \frac{2nK\,|\mathcal{F}|}{\delta}\right)}$$

*Proof of Lemma 9.* We will show the following claim: for every node v in $\mathcal{T}$, there exists an event $E_{\mathtt{v}}$ that happens with probability at least $1 - \delta|\mathcal{T}^{\mathtt{v}}|/2K$, in which

$$\mathbb{E}\left[\mathbb{E}[\ell_h(\mathcal{T}^{\mathtt{v}}(x)) \mid x] - \min_{a \in \text{range}(\mathcal{T}^{\mathtt{v}})} \mathbb{E}[\ell_h(a) \mid x]\right] = \mathbb{E}\left[\ell_h(\mathcal{T}^{\mathtt{v}}(x))\right] - \mathbb{E}\left[\min_{a \in \text{range}(\mathcal{T}^{\mathtt{v}})} \mathbb{E}[\ell_h(a) \mid x]\right]$$

$$\leq |\mathcal{T}^{\mathtt{v}}|\left(8\sqrt{\frac{\ln \frac{2n'K|\mathcal{F}|}{\delta}}{n'p_{\min}h}} + 4\frac{\ln \frac{2n'K|\mathcal{F}|}{\delta}}{n'p_{\min}h}\right), \quad (2)$$

where $|\mathcal{T}^{\mathtt{v}}|$ is the total number of nodes in subtree $\mathcal{T}^{\mathtt{v}}$ (including internal nodes and leaves), and $n' = \frac{n}{\log K}$ is the number of examples for training at each level of $\mathcal{T}$. As $\mathcal{T}$ is a complete binary tree with $K - 1$ internal nodes and $K$ leaves, $|\mathcal{T}^{\mathtt{v}}| = 2K - 1$. To see why it completes the proof, we set v to be the root of $\mathcal{T}$. In this case, we get that with probability $1 - \delta|\mathcal{T}|/2K \geq 1 - \delta$,

$$\mathbb{E}\left[\ell_h(\mathcal{T}(x))\right] - \mathbb{E}\left[\min_{a \in \mathcal{A}_K} \mathbb{E}[\ell_h(a) \mid x]\right] \leq (2K - 1) \cdot \left(8\sqrt{\frac{\ln \frac{2n'K|\mathcal{F}|}{\delta}}{n'p_{\min}h}} + 4\frac{\ln \frac{2n'K|\mathcal{F}|}{\delta}}{n'p_{\min}h}\right).$$

Observing that as $\text{range}(\mathcal{T}') = \mathcal{A}_K$ for all $\mathcal{T}'$ in $\mathcal{F}_K$, we have that $\mathbb{E}\left[\min_{a \in \mathcal{A}_K} \mathbb{E}[\ell_h(a) \mid x]\right] \leq \min_{\mathcal{T}' \in \mathcal{F}_K} \mathbb{E}\left[\mathbb{E}[\ell_h(\mathcal{T}'(x)) \mid x]\right] = \min_{\mathcal{T}' \in \mathcal{F}_K} V(\mathcal{T}_h)$. In conjunction with the fact that $\frac{n}{2\log K} \leq n' \leq n$, we get that

$$\mathbb{E}\left[\ell_h(\mathcal{T}(x))\right] - \min_{\mathcal{T}' \in \mathcal{F}_K} \mathbb{E}\left[\ell_h(\mathcal{T}'(x))\right] \leq \left(16\sqrt{\frac{K^2 \log K \ln \frac{2nK|\mathcal{F}|}{\delta}}{np_{\min}h}} + \frac{8K \log K \ln \frac{2nK|\mathcal{F}|}{\delta}}{np_{\min}h}\right).$$

The lemma follows, because if $\frac{8K \log K \ln \frac{2nK|\mathcal{F}|}{\delta}}{np_{\min}h} \geq \frac{1}{2}$, the lemma statement is trivially true, as the right hand is at least 1, and the left hand side is at most 1; otherwise, $\frac{8K \log K \ln \frac{2nK|\mathcal{F}|}{\delta}}{np_{\min}h} \leq 4\sqrt{\frac{K \log K \ln \frac{2nK|\mathcal{F}|}{\delta}}{np_{\min}h}}$, in which case the right hand side is at most $20\sqrt{\frac{K^2 \log K \ln \frac{2nK|\mathcal{F}|}{\delta}}{np_{\min}h}}$.

Next we turn to show the above claim by induction.

**Base case.** If v is of depth $D - 1$, i.e. it is the parent of a pair of leaves $\mathtt{l} \triangleq \mathtt{v.left} \in \mathcal{A}_K$ and $\mathtt{r} \triangleq \mathtt{v.right} \in \mathcal{A}_K$, then $c^{\mathtt{v}}(\text{left}) = \tilde{c}(\text{label}(\mathtt{l}))$, $c^{\mathtt{v}}(\text{right}) = \tilde{c}(\text{label}(\mathtt{r}))$. In addition, $\text{range}(\mathcal{T}^{\mathtt{v}}) = \{\text{label}(\mathtt{l}), \text{label}(\mathtt{r})\}$. Given a classifier $f : \mathcal{X} \to \{\text{left}, \text{right}\}$ in $\mathcal{F}$, we define its induced tree policy at node v, $\pi_f : \mathcal{X} \to \mathcal{A}_K$, as: $\pi_f(x) = \text{label}(\mathtt{v}.f(x))$.

Observe that the CSMC examples $\left\{(x_s, \tilde{c}_s^h)\right\}_{s \in B_{D-1}}$ (where $B_{D-1} = [n']$) can be viewed as induced by a set of $p_{\min}$-well-formed logged data $\{(x_s, a_s, P_s(a_s \mid x_s), \ell_s(a_s))\}_{s \in B_{D-1}}$ using IPS. From Lemma 16 in Appendix E, we have that there exists an event $E_{\mathtt{v}}$ such that $\mathbb{P}(E_{\mathtt{v}}) \geq 1 - \delta/K$, on which for all $f$ in $\mathcal{F}$,

$$\left|\hat{V}_{B_{H-1}}(\pi_{f,h}) - V(\pi_{f,h})\right| \leq \left(4\sqrt{\frac{\ln |\mathcal{F}| + \ln \frac{2n'K}{\delta}}{n'p_{\min}h}} + 2\frac{\ln |\mathcal{F}| + \ln \frac{2n'K}{\delta}}{n'p_{\min}h}\right). \quad (3)$$

We henceforth condition on $E_{\mathtt{v}}$ happening.

Observe that $\hat{V}_{B_{D-1}}(\pi_{f,h}) = \mathbb{E}_{S^{\mathtt{v}}}[c^{\mathtt{v}}(f(x))]$; As $f^{\mathtt{v}} = \text{argmin}_{f \in \mathcal{F}} \mathbb{E}_{S^{\mathtt{v}}}[c^{\mathtt{v}}(f(x))]$, we have that: $\hat{V}_{B_{D-1}}(\pi_{f^{\mathtt{v}},h}) \leq \hat{V}_{B_{D-1}}(\pi_{f^{\mathtt{v},\star},h})$ for $f^{\mathtt{v},\star}$ defined in Definition 5. This fact, in conjunction with Equation (3), gives that

$$V(\pi_{f^{\mathtt{v}},h}) - V(\pi_{f^{\mathtt{v},\star},h}) \leq \left(8\sqrt{\frac{\ln |\mathcal{F}| + \ln \frac{2n'K}{\delta}}{n'p_{\min}h}} + 4\frac{\ln |\mathcal{F}| + \ln \frac{2n'K}{\delta}}{n'p_{\min}h}\right).$$

Also, by Definition 5, $V(\pi_{f^{\mathtt{v},\star},h}) = \mathbb{E}\left[\ell_h(\pi_{f^{\mathtt{v},\star}}(x))\right] = \mathbb{E}\left[\ell_h(\text{label}(\mathtt{v}.f^{\mathtt{v},\star}(x)))\right] = \mathbb{E}\left[\min_{a \in \text{range}(\mathcal{T}^{\mathtt{v}})} \mathbb{E}[c(a) \mid x]\right]$. In addition, by the definition of $\pi_f$, $\mathcal{T}^{\mathtt{v}} = \pi_{f^{\mathtt{v}}}$. Therefore,

$$\mathbb{E}[\ell_h(\mathcal{T}^{\mathtt{v}}(x))] - \mathbb{E}\left[\min_{a \in \text{range}(\mathcal{T}^{\mathtt{v}})} \mathbb{E}[\ell_h(a) \mid x]\right] \leq \left(8\sqrt{\frac{\ln |\mathcal{F}| + \ln \frac{2n'K}{\delta}}{n'p_{\min}h}} + 4\frac{\ln |\mathcal{F}| + \ln \frac{2n'K}{\delta}}{n'p_{\min}h}\right),$$

proving the base case.

**Inductive case.** Suppose that the results holds for all nodes $\mathtt{v}$ at level $\geq d+1$. For node $\mathtt{v}$ at depth $d$, suppose $\mathtt{l} = \mathtt{v.left}$ and $\mathtt{r} = \mathtt{v.right}$ are its two children at level $d+1$. In this notation, given an IPS CSMC example $(x, \tilde{c})$ in $B_d$, $c^{\mathtt{v}}(\mathtt{left}) = \tilde{c}(\mathcal{T}^{\mathtt{l}}(x))$, $c^{\mathtt{v}}(\mathtt{right}) = \tilde{c}(\mathcal{T}^{\mathtt{r}}(x))$. Given a classifier $f : \mathcal{X} \to \{\mathtt{left}, \mathtt{right}\}$ in $\mathcal{F}$, and the subtree policies $\mathcal{T}^{\mathtt{l}}, \mathcal{T}^{\mathtt{r}}$, we define its induced tree policy at $\mathtt{v}$, $\pi_f : \mathcal{X} \to \mathcal{A}_K$ as: $\pi_f(x) = \mathcal{T}^{\mathtt{v}.f(x)}(x)$.

First, consider the training of classifier $f^{\mathtt{v}}$ at node $\mathtt{v}$. We note that given logged data with indices $\cup_{d'=d+1}^{H-1} B_{d'} = [(H-d-1)n']$ used to learn downstream classifiers in internal nodes of $\mathcal{T}_{\mathtt{l}}$ and $\mathcal{T}_{\mathtt{r}}$, the CSMC examples $\left\{(x_s, \tilde{c}_s^h)\right\}_{s \in B_l}$ can be viewed as induced by a set of $p_{\min}$-well-formed logged data $\{(x_s, a_s, P_s(a_s \mid x_s), \ell_s(a_s))\}_{s \in B_d}$ using IPS (See Definition 8). Therefore, applying Lemma 16, we get that there exists an event $E_{\mathtt{v}}^1$ such that $\mathbb{P}(E_{\mathtt{v}}^1) \geq 1 - \delta/K$, on which for all $f$ in $\mathcal{F}$,

$$\left|\hat{V}_{B_d}(\pi_{f,h}) - V(\pi_{f,h})\right| \leq \left(4\sqrt{\frac{\ln|\mathcal{F}| + \ln\frac{2n'K}{\delta}}{n'p_{\min}h}} + 2\frac{\ln|\mathcal{F}| + \ln\frac{2n'K}{\delta}}{n'p_{\min}h}\right). \tag{4}$$

In addition, by inductive hypothesis, we have that there exists two events $E_{\mathtt{l}}$ and $E_{\mathtt{r}}$, happening with probability $1 - |\mathcal{T}_{\mathtt{l}}|\delta/2K$ and $1 - |\mathcal{T}_{\mathtt{r}}|\delta/2K$ respectively, in which

$$\mathbb{E}\left[\mathbb{E}[\ell_h(\mathcal{T}^{\mathtt{l}}(x))]|x] - \min_{a \in \mathrm{range}(\mathcal{T}^{\mathtt{l}})} \mathbb{E}[\ell_h(a)|x]\right]$$
$$\leq |\mathcal{T}_{\mathtt{l}}|\left(8\sqrt{\frac{\ln|\mathcal{F}| + \ln\frac{2n'K}{\delta}}{n'p_{\min}h}} + 4\frac{\ln|\mathcal{F}| + \ln\frac{2n'K}{\delta}}{n'p_{\min}h}\right), \tag{5}$$

and

$$\mathbb{E}\left[\mathbb{E}[\ell_h(\mathcal{T}^{\mathtt{r}}(x))]|x] - \min_{a \in \mathrm{range}(\mathcal{T}^{\mathtt{r}})} \mathbb{E}[\ell_h(a)|x]\right]$$
$$\leq |\mathcal{T}_{\mathtt{r}}|\left(8\sqrt{\frac{\ln|\mathcal{F}| + \ln\frac{2n'K}{\delta}}{n'p_{\min}h}} + 4\frac{\ln|\mathcal{F}| + \ln\frac{2n'K}{\delta}}{n'p_{\min}h}\right), \tag{6}$$

holds respectively. We define $E_{\mathtt{v}} = E_{\mathtt{v}}^1 \cap E_{\mathtt{l}} \cap E_{\mathtt{r}}$. By union bound, $\mathbb{P}(E_{\mathtt{v}}) \geq 1 - |\mathcal{T}_{\mathtt{v}}|/2K$. We henceforth condition on $E_{\mathtt{v}}$ happening.

First, we note that by Equation (4) and the optimality of $\pi_{f^{\mathtt{v}},h}$, $\hat{V}_{B_d}(\pi_{f^{\mathtt{v}},h}) \leq \hat{V}_{B_d}(\pi_{f^{\mathtt{v},\star},h})$ for $f^{\mathtt{v},\star}$ defined in Definition 5. This fact, in conjunction with Equation (4), gives that

$$\mathbb{E}\left[\ell_h(\mathcal{T}^{\mathtt{v}}(x))\right] - \mathbb{E}\left[\ell_h(\mathcal{T}^{\mathtt{v}.f^{\mathtt{v},\star}(x)}(x))\right] = V(\pi_{f^{\mathtt{v}},h}) - V(\pi_{f^{\mathtt{v},\star},h})$$
$$\leq \left(8\sqrt{\frac{\ln|\mathcal{F}| + \ln\frac{2n'K}{\delta}}{n'p_{\min}h}} + 4\frac{\ln|\mathcal{F}| + \ln\frac{2n'K}{\delta}}{n'p_{\min}h}\right). \tag{7}$$

We have the following inequalities:

$$\mathbb{E}\left[\ell_h(\mathcal{T}^{\mathtt{v}.f^{\mathtt{v},\star}(x)}(x))\right]$$
$$= \mathbb{E}\left[\mathbb{E}[\ell_h(\mathcal{T}^{\mathtt{l}}(x))|x]\,\mathbb{I}(f^{\mathtt{v},\star}(x) = \mathtt{left}) + \mathbb{E}[\ell_h(\mathcal{T}^{\mathtt{r}}(x))|x]\,\mathbb{I}(f^{\mathtt{v},\star}(x) = \mathtt{right})\right]$$
$$\leq \mathbb{E}\left[\min_{a \in \mathrm{range}(\mathcal{T}^{\mathtt{l}})} \mathbb{E}[c(a)|x]\,\mathbb{I}(f^{\mathtt{v},\star}(x) = \mathtt{left})\right] + \mathbb{E}\left[\min_{a \in \mathrm{range}(\mathcal{T}^{\mathtt{r}})} \mathbb{E}[c(a)|x]\,\mathbb{I}(f^{\mathtt{v},\star}(x) = \mathtt{right})\right]$$
$$\quad + (|\mathcal{T}^{\mathtt{l}}| + |\mathcal{T}^{\mathtt{r}}|)\left(8\sqrt{\frac{\ln|\mathcal{F}| + \ln\frac{n'K}{\delta}}{n'p_{\min}h}} + 4\frac{\ln|\mathcal{F}| + \ln\frac{n'K}{\delta}}{n'p_{\min}h}\right)$$
$$\leq \mathbb{E}\left[\min_{a \in \mathrm{range}(\mathcal{T}^{\mathtt{v}})} \mathbb{E}[c(a)|x]\right] + (|\mathcal{T}^{\mathtt{l}}| + |\mathcal{T}^{\mathtt{r}}|)\left(8\sqrt{\frac{\ln|\mathcal{F}| + \ln\frac{2n'K}{\delta}}{n'p_{\min}h}} + 4\frac{\ln|\mathcal{F}| + \ln\frac{2n'K}{\delta}}{n'p_{\min}h}\right). \tag{8}$$

where the first inequality is from Equations (5) and (6), the second inequality is from the $(h, K)$-realizability assumption.

Therefore, combining Equations (7) and (8), we get

$$\mathbb{E}\left[\ell_h(\mathcal{T}^{\mathtt{v}}(x))\right] - \mathbb{E}\left[\min_{a \in \mathrm{range}(\mathcal{T}^v)} \mathbb{E}[\ell_h(a)|x]\right]$$

$$\leq (1 + |\mathcal{T}^1| + |\mathcal{T}^{\mathtt{r}}|)\left(8\sqrt{\frac{\ln|\mathcal{F}| + \ln\frac{2n'K}{\delta}}{n'p_{\min}h}} + 4\frac{\ln|\mathcal{F}| + \ln\frac{2n'K}{\delta}}{n'p_{\min}h}\right)$$

$$= |\mathcal{T}^{\mathtt{v}}|\left(8\sqrt{\frac{\ln|\mathcal{F}| + \ln\frac{2n'K}{\delta}}{n'p_{\min}h}} + 4\frac{\ln|\mathcal{F}| + \ln\frac{2n'K}{\delta}}{n'p_{\min}h}\right).$$

This completes the induction, and proves the claim. □

## B.2 Proof of Theorem 6

We first give a formal statement of Theorem 6 in the theorem below.

**Theorem 10.** *Suppose Algorithm 1 is run with greedy parameter $\epsilon$, smoothing parameter $h$, discretization scale $K$, and base hypothesis class $\mathcal{F}$. In addition, suppose $(\mathcal{F}, \mathcal{D})$ is $(h, K)$-realizable. Then with probability $1 - \delta$, it has $h$-smoothed regret against $\mathcal{F}_\infty$ bounded as:*

$$\mathrm{Reg}(T, \mathcal{F}_\infty, h) \leq \mathcal{O}\left(\left(\epsilon + \frac{1}{Kh}\right)T + K\sqrt{\frac{T}{\epsilon h} \cdot \left(\ln\frac{|\mathcal{F}|}{\delta}\right)}\right).$$

*Taking $\epsilon = \left(\frac{\ln\frac{|\mathcal{F}|}{\delta}}{Th^3}\right)^{1/5}$, $K = \left(\frac{T}{h^2 \ln\frac{|\mathcal{F}|}{\delta}}\right)^{1/5}$, we have $\mathrm{Reg}(T, \mathcal{F}_\infty, h) \leq \mathcal{O}\left(\left(T^4 \ln\frac{|\mathcal{F}|}{\delta}/h^3\right)^{1/5}\right)$.*

*Proof of Theorem 10.* We will show that with probability $1 - \delta$,

$$\mathrm{Reg}(T, \mathcal{F}_K, h) \leq \mathcal{O}\left(\epsilon T + K\sqrt{\frac{T}{\epsilon h} \cdot \left(\ln\frac{2TK|\mathcal{F}|}{\delta}\right)}\right),$$

to see why this completes the proof, we observe that for any policy $\mathcal{T}$ in $\mathcal{F}_\infty$, there is a policy $\mathcal{T}_K$ in $\mathcal{F}_K$, such that $|\mathcal{T}_K(x) - \mathcal{T}(x)| \leq \frac{1}{K}$: we can take $\mathcal{T}_K$ to be a truncation of $\mathcal{T}$ that only keeps its top $\log K$ levels. In addition, as $\ell_h$ is $1/h$-Lipschitz, we have

$$\mathbb{E}\left[\ell_h(\mathcal{T}_K(x))\right] \leq \mathbb{E}\left[\ell_h(\mathcal{T}(x))\right] + \frac{1}{Kh}.$$

This implies that $\min_{\mathcal{T} \in \mathcal{F}_K} \mathbb{E}\left[\ell_h(\mathcal{T}(x))\right] \leq \min_{\mathcal{T}' \in \mathcal{F}_\infty} \mathbb{E}\left[\ell_h(\mathcal{T}'(x))\right] + \frac{1}{Kh}$. As a result,

$$\mathrm{Reg}(T, \mathcal{F}_\infty, h) \leq \mathrm{Reg}(T, \mathcal{F}_K, h) + \frac{T}{Kh} = \mathcal{O}\left(\left(\epsilon + \frac{1}{Kh}\right)T + K\sqrt{\frac{T}{\epsilon h} \cdot \left(\ln\frac{2TK|\mathcal{F}|}{\delta}\right)}\right).$$

We now come back to the proof of the above claim. First observe that the $h$-smoothed regret can be rewritten as:

$$\mathrm{Reg}(T, \mathcal{F}_K, h) = \sum_{t=1}^{T}\left(\mathbb{E}\left[\ell_t(a_t)\right] - \min_{\mathcal{T}' \in \mathcal{F}_K} V(\mathcal{T}'_h)\right). \tag{9}$$

Let $\pi_{t+1}$ denote the tree $\mathcal{T}$ at the beginning of time step $t+1$, which is learned from CSMC examples $\{(x_s, \tilde{c}_s)\}_{s=1}^{t}$ by $\mathtt{Train\_tree}$. Define event

$$E = \left\{\text{for all time steps } t \text{ in } [T-1], V(\pi_{t+1,h}) \leq \min_{\mathcal{T}' \in \mathcal{F}_K} V(\mathcal{T}'_h) + 20\sqrt{\frac{K^2 \log K}{\epsilon h t} \cdot \left(\ln\frac{2TK|\mathcal{F}|}{\delta}\right)}\right\}.$$

From Lemma 9 with $p_{\min} = \epsilon$, $\delta' = \frac{\delta}{T}$, and a union bound over all $t \in [T]$, we get that $\mathbb{P}(E) \geq 1-\delta$.

Now, conditioned on event $E$ happening, we conclude the regret bound. We first have the following upper bound on the algorithm's instantaneous loss at time $t$, namely $\mathbb{E}\left[\ell_t(a_t)\right]$:

$$\mathbb{E}[\ell_t(a_t)] = (1 - \epsilon) \cdot \mathbb{E}_{(x_t, \ell_t) \sim D} \mathbb{E}_{a \sim \pi_{t,h}(\cdot | x_t)}[\ell_t(a)] + \epsilon \cdot \mathbb{E}_{(x_t, \ell_t) \sim D} \mathbb{E}_{a \sim U(\mathcal{A})}[\ell_t(a)]$$
$$\leq V(\pi_{t,h}) + \epsilon. \tag{10}$$

Therefore, for all $t \in \{2, \ldots, T\}$, we have

$$\mathbb{E}[\ell_t(a_t)] \leq \epsilon + \min_{\mathcal{T}' \in \mathcal{F}_K} V(\mathcal{T}_h') + 20\sqrt{\frac{K^2 \log K}{\epsilon h(t-1)} \cdot \left(\ln \frac{2TK|\mathcal{F}|}{\delta}\right)} \tag{11}$$

We now conclude the regret bound:

$$\text{Reg}(\mathcal{F}_K, T, h) = \sum_{t=1}^{T} \left( \mathbb{E}[\ell_t(a_t)] - \min_{\mathcal{T}' \in \mathcal{F}_K} V(\mathcal{T}_h') \right)$$

$$\leq 1 + \epsilon(T-1) + \sum_{t=2}^{T} 20\sqrt{\frac{K^2 \log K}{\epsilon h(t-1)} \cdot \left(\ln \frac{2TK|\mathcal{F}|}{\delta}\right)}$$

$$\leq 1 + \epsilon T + 40 \cdot \sqrt{\frac{TK^2 \log K}{\epsilon h} \cdot \left(\ln \frac{2TK|\mathcal{F}|}{\delta}\right)}.$$

where the first inequality uses the fact that $\mathbb{E}[\ell_t(a_t)] - \min_{\mathcal{T}' \in \mathcal{F}_K} V(\mathcal{T}_h')$ is at most 1 if $t = 1$, and is at most $\epsilon + 20\sqrt{\frac{K^2 \log K}{\epsilon h(t-1)} \cdot \left(\ln \frac{2TK|\mathcal{F}|}{\delta}\right)}$ if $t \geq 2$, and the second inequality uses the fact that $\sum_{t=1}^{T-1} \frac{1}{\sqrt{t}} \leq 2\sqrt{T}$. The theorem follows. $\qquad \square$

## B.3 Proof of Theorem 7

We first give a formal statement of Theorem 7 below.

**Theorem 11.** *Suppose Algorithm 3 is run with a set of $p_{\min}$-well-formed logged data $\{x_t, a_t, P_t(a_t \mid x_t), \ell_t(a_t)\}_{t=1}^{T}$, set of (bandwidth, discretization) combinations $\mathcal{J} \subset [0,1] \times 2^{\mathbb{N}}$, base hypothesis class $\mathcal{F}$. In addition, suppose $(\mathcal{F}, \mathcal{D})$ is $(h, K)$-realizable for all $(h, K) \in \mathcal{J}$. Then, with probability $1 - \delta$, its returned policy $\hat{\pi}$ ensures:*

$$\lambda_0(\hat{\pi}) \leq \min_{(h,K) \in \mathcal{J}, \pi \in \mathcal{F}_K} \left( \lambda_h(\pi) + \mathcal{O}\left( K \sqrt{\ln \frac{|\mathcal{F}||\mathcal{J}|}{\delta} / (p_{\min} hT)} \right) \right).$$

*Proof of Theorem 11.* For every $(h, K)$ in $\mathcal{J}$, recall that $\mathcal{T}_t^{h,K}$ denotes the policy trained by CATS_Off at the beginning of iteration $t$ for that $(h, K)$ combination.

Define events

$$E_1 = \Big\{ \forall (h, K) \in \mathcal{J}, \forall t \in [T-1], V(\mathcal{T}_{t+1,h}^{h,K}) \leq \min_{\mathcal{T}' \in \mathcal{F}_K} V(\mathcal{T}_h')$$
$$+ 20\sqrt{\frac{K^2 \log K}{p_{\min} ht} \cdot \left(\ln \frac{4TK|\mathcal{F}||\mathcal{J}|}{\delta}\right)} \Big\}$$

$$E_2 = \Big\{ \forall (h, K) \in \mathcal{J}, \forall t \in [T], \left| \frac{1}{T} \sum_{t=1}^{T} \tilde{c}_t^h(\mathcal{T}_t^{h,K}(x_t)) - \frac{1}{T} \sum_{t=1}^{T} V(\mathcal{T}_{t,h}^{h,K}) \right|$$
$$\leq 8\sqrt{\left(\frac{1}{T}\sum_{t=1}^{T} V(\mathcal{T}_{t,h}^{h,K})\right) \cdot \frac{\ln \frac{2T|\mathcal{J}|}{\delta}}{p_{\min} hT} + 4 \frac{\ln \frac{4T|\mathcal{J}|}{\delta}}{p_{\min} hT}} \Big\}$$

From Lemma 9 in Appendix E and union bound, we know that $\mathbb{P}(E_1) \geq 1 - \frac{\delta}{2}$; from Lemma 16, item 1 and union bound over all $(h, K) \in \mathcal{J}$, we get that $\mathbb{P}(E_2) \geq 1 - \frac{\delta}{2}$. Define event $E \triangleq E_1 \cap E_2$. By union bound, $\mathbb{P}(E) \geq 1 - \delta$. We henceforth condition on event $E$ happening.

We denote $\hat{g}(h, K) \triangleq \frac{1}{T} \sum_{t=1}^{T} \tilde{c}_t^h(\mathcal{T}_t^{h,K}(x_t))$, $g(h, K) \triangleq \frac{1}{T} \sum_{t=1}^{T} V(\mathcal{T}_{t,h}^{h,K})$, $\sigma(h, K) \triangleq \frac{64 \ln \frac{4T|\mathcal{J}|}{\delta}}{p_{\min} hT}$. Using this notation, and by the definition of $E_2$, for all $(h, K)$ in $\mathcal{J}$,

$$|\hat{g}(h, K) - g(h, K)| \leq \sqrt{g(h, K)\sigma(h, K)} + \sigma(h, K)$$

Specifically,

$$g(h, K) \leq \hat{g}(h, K) + \sqrt{\hat{g}(h, K)\sigma(h, K)} + \sigma(h, K), \tag{12}$$

In addition, from the elementary fact that $A \leq B + C\sqrt{A} \Rightarrow A \leq B + C^2 + C\sqrt{B}$, we have

$$g(h, K) \leq \hat{g}(h, K) + \sqrt{\hat{g}(h, K)\sigma(h, K)} + 3\sigma(h, K). \tag{13}$$

By the optimality of $\hat{h}, \hat{K}$, for all $(h, K)$ in $\mathcal{J}$,

$$\hat{g}(\hat{h}, \hat{K}) + \sqrt{\hat{g}(\hat{h}, \hat{K})\sigma(\hat{h}, \hat{K})} + 3\sigma(\hat{h}, \hat{K}) \leq \hat{g}(h, K) + \sqrt{\hat{g}(h, K)\sigma(h, K)} + 3\sigma(h, K). \tag{14}$$

Therefore, we have the following set of inequalities for every $h \in \mathcal{H}$ and $K \in \mathcal{K}$:

$$
\begin{aligned}
g(\hat{h}, \hat{K}) &\leq \hat{g}(\hat{h}, \hat{K}) + \sqrt{\hat{g}(\hat{h}, \hat{K})\sigma(\hat{h}, \hat{K})} + 3\sigma(\hat{h}, \hat{K}) \\
&\leq \hat{g}(h, K) + \sqrt{\hat{g}(h, K)\sigma(h, K)} + 3\sigma(h, K) \\
&\leq g(h, K) + 3\sqrt{g(h, K)\sigma(h, K)} + 6\sigma(h, K)
\end{aligned}
\tag{15}
$$

where the first inequality uses Equation (12); the second inequality is from Equation (14), the third inequality again uses Equation (12) and algebra.

We claim that $g(\hat{h}, \hat{K}) \leq g(h, K) + 9\sqrt{\sigma(h, K)}$, because If $\sigma(h, K) \geq 1$, the statement is trivially true as $g(\hat{h}, \hat{K}) \leq 1$; otherwise, $6\sigma(h, K) \leq 6\sqrt{\sigma(h, K)}$, and the RHS of the above inequality is at most $g(h, K) + (3 + 6)\sqrt{\sigma(h, K)} \leq g(h, K) + 9\sqrt{\sigma(h, K)}$.

Rephrasing the above inequality using our previous notation, we have:

$$\frac{1}{T} \sum_{t=1}^{T} V(\mathcal{T}_{t,\hat{h}}^{\hat{h}, \hat{K}}) \leq \frac{1}{T} \sum_{t=1}^{T} V(\mathcal{T}_{t,h}^{h,K}) + 72\sqrt{\frac{\ln \frac{4T|\mathcal{J}|}{\delta}}{p_{\min} hT}}. \tag{16}$$

Meanwhile, observe that by the definition of $E_1$, we can bound $\frac{1}{T} \sum_{t=1}^{T} V(\mathcal{T}_{t-1,h}^{h,K})$ as follows:

$$
\begin{aligned}
\frac{1}{T} \sum_{t=1}^{T} V(\mathcal{T}_{t,h}^{h,K}) &\leq \min_{\mathcal{T} \in \mathcal{F}_K} V(\mathcal{T}_h) + \frac{1}{T}\left(1 + \sum_{t=1}^{T-1} 44\sqrt{\frac{K^2 \log K}{p_{\min} ht} \cdot \left(\ln \frac{4TK|\mathcal{F}||\mathcal{J}|}{\delta}\right)}\right) \\
&\leq \min_{\mathcal{T} \in \mathcal{F}_K} V(\mathcal{T}_h) + \frac{1}{T} + 88\sqrt{\frac{K^2 \log K}{p_{\min} hT} \cdot \left(\ln \frac{4TK|\mathcal{F}||\mathcal{J}|}{\delta}\right)}
\end{aligned}
\tag{17}
$$

where the first inequality uses the simple fact that $V(\mathcal{T}_{0,h}^{h,K}) \leq 1$; the second inequality uses the algebraic fact that $\sum_{t=1}^{T-1} \frac{1}{\sqrt{t}} \leq 2\sqrt{T}$.

Combining Equations (16) and (17), along with some algebra, we get:

$$
\begin{aligned}
\frac{1}{T} \sum_{t=1}^{T} V(\mathcal{T}_{t,\hat{h}}^{\hat{h}, \hat{K}}) &\leq \min_{\mathcal{T} \in \mathcal{F}_K} V(\mathcal{T}_h) + \frac{1}{T} + 88\sqrt{\frac{K^2 \log K}{p_{\min} hT} \cdot \left(\ln \frac{4TK|\mathcal{F}||\mathcal{J}|}{\delta}\right)} + 72\sqrt{\frac{\ln \frac{4T|\mathcal{J}|}{\delta}}{p_{\min} hT}} \\
&\leq \min_{\mathcal{T} \in \mathcal{F}_K} V(\mathcal{T}_h) + 160\sqrt{\frac{K^2 \log K}{p_{\min} hT} \cdot \left(\ln \frac{4TK|\mathcal{F}||\mathcal{J}|}{\delta}\right)}.
\end{aligned}
$$

The theorem follows by recognizing that the left hand side is $\mathbb{E}V(\hat{\pi}) = \mathbb{E}\lambda_0(\hat{\pi})$, where $\hat{\pi}$ is drawn uniformly at random from $\left\{\mathcal{T}_{t,\hat{h}}^{\hat{h}, \hat{K}}\right\}_{t=1}^{T}$. $\qquad \square$

## C   `CATS` **with adaptive bandwidth**

As can be seen from Theorem 6, `CATS` obtains smoothed regret guarantees with respect to a fixed value of $h$; in practice, as different loss function have different smoothness properties, it would be useful to develop an algorithm that has performance competitive with $\mathcal{T}_h$ for all $\mathcal{T}$ in $\mathcal{F}_\infty$ and all $h$ in $(0,1]$ simultaneously. In this section, we develop a variant of `CATS`, namely Algorithm 5, that has such guarantees. Specifically, with appropriate tuning of its greedy parameters, it achieves the following type of high-probability regret guarantee for some function $R$ in terms of bandwidth $h$, number of discretized actions $K$, base class $\mathcal{F}$, time horizon $T$:

$$\forall h \in [0,1] \text{. } \mathrm{Reg}(T, \mathcal{F}_\infty, h) \le R(h, K, |\mathcal{F}|, T),$$

under the realizability assumptions stated in Definition 5.

At a high level, Algorithm 5 follows the same outline of Algorithm 1: it has an $\epsilon$-greedy action selection step (lines 4 to 5) and has a tree training step (lines 6 to 8). A crucial difference between Algorithm 5 and Algorithm 1 is that, it now maintains $|\mathcal{H}|$ policies $\left\{\mathcal{T}_t^h\right\}_{h \in \mathcal{H}}$ over time as opposed to only one; to this end, it accumulates $|\mathcal{H}|$ CSMC datasets $\left\{\left\{(x_s, \tilde{c}_s^h)\right\}_{s=1}^t\right\}_{h \in \mathcal{H}}$. After generating policies $\left\{\mathcal{T}_t^h\right\}_{h \in \mathcal{H}}$, it selects $\mathcal{T}_t^{h_t}$ using structural risk minimization [55] (line 9). This choice of $h_t$ ensures that the expected loss of $\mathcal{T}_{t,h_t}^{h_t}$ is competitive with all $\mathcal{T}_{t,h}^h$'s. Finally, we remark that the set of bandwidth $\mathcal{H}$ acts as a covering of the $[0,1]$ interval; as we will see, setting $\mathcal{H}$ to be a fine grid as in Algorithm 5 ensures that for any $\mathcal{T}$ in $\mathcal{F}_K$, and every $h$ in $[0,1]$, there exists a $h'$ in $\mathcal{H}$ such that the optimal $\mathcal{T}_{h'}$ has expected loss close to that of $\mathcal{T}_h$.

---

**Algorithm 5** `CATS` with adaptive bandwidth

---

**Input:** Greedy parameter $\epsilon$, number of discretized actions $K = 2^D$, base class $\mathcal{F}$.
1: Let $\mathcal{H} = \left\{h \in \left\{\frac{1}{4T^2}, \frac{2}{4T^2}, \ldots, 1\right\} : h \ge \frac{1}{2T}\right\}$ be the set of bandwidths in consideration.
2: Let $\pi_t$ be an arbitrary policy in $\mathcal{F}_K$.
3: **for** $t = 1, 2, \ldots, T$ **do**
4:     Define policy $P_t(a \mid x) := (1 - \epsilon)\pi_t(a|x) + \epsilon$.
5:     Observe context $x_t$, select action $a_t \sim P_t(\cdot \mid x_t)$, observe cost $\ell_t(a_t)$.
6:     **for** all $h$ in $\mathcal{H}$ **do**
7:         $\tilde{c}_t^h(i/K) \leftarrow \frac{\mathtt{Smooth}_h(a_t|i/K)}{P_t(a_t|x_t)}\ell_t(a_t)$ for all $i$.
8:         Let $\mathcal{T}^h \leftarrow \mathtt{Train\_tree}\left(\left\{(x_s, \tilde{c}_s^h)\right\}_{s=1}^t\right)$.
9:     Let $h_t \in \mathrm{argmin}_{h \in \mathcal{H}} \left(\hat{V}_t(\mathcal{T}_h^h) + 4\sqrt{\frac{K \ln |\mathcal{F}| + \ln \frac{8T^4}{\delta}}{t\epsilon h}} + 2\frac{K \ln |\mathcal{F}| + \ln \frac{8T^4}{\delta}}{t\epsilon h}\right)$, and let $\pi_{t+1} = \mathcal{T}_{h_t}^{h_t}$.

---

We next present a theorem on the regret guarantee of Algorithm 5.

**Theorem 12.** *Suppose Algorithm 5 is run with greedy parameter $\epsilon$, number of discretized actions $K$, and base class $\mathcal{F}$. In addition, suppose $(\mathcal{F}, \mathcal{D})$ satisfies the $(h, K)$-realizability assumption for all $h \in (0,1)$. Then with probability $1 - \delta$, it has uniform $h$-smoothed regret bounded as:*

$$\forall h \in [0,1] \text{. } \mathrm{Reg}(T, \mathcal{F}_\infty, h) \le \tilde{\mathcal{O}}\left((\epsilon + \frac{1}{Kh})T + \sqrt{\frac{K^2 \log K \cdot T \cdot \left(\ln \frac{|\mathcal{F}|}{\delta}\right)}{\epsilon h}}\right).$$

*Specifically, by taking $\epsilon = \left(\frac{\ln \frac{|\mathcal{F}|}{\delta}}{T}\right)^{1/5}$, $K = \left(\frac{T}{\ln \frac{|\mathcal{F}|}{\delta}}\right)^{1/5}$, we have*

$$\forall h \in [0,1] \text{. } \mathrm{Reg}(T, \mathcal{F}_\infty, h) \le \tilde{\mathcal{O}}\left(\frac{1}{h} \cdot \left(T^4 \ln \frac{|\mathcal{F}|}{\delta}\right)^{1/5}\right).$$

Before going into the proof of the theorem, we remark that the only difference between the above regret guarantee of Algorithm 5 and that of CATS (Theorem 6) is that, the order of $h$ is different ($\frac{1}{h}$ versus $\frac{1}{h^{3/5}}$). This can be seen as a price we pay for adaptivity: Algorithm 5 sets $K$ independent of $h$, whereas CATS can set $K$ that depends on $h$.

*Proof sketch.* By standard analysis on structural risk minimization [see e.g. 55], and union bound, it can be shown that with probability $1 - \delta/2$, for all time steps $t$ in $[T]$ and all $h \in \mathcal{H}$,

$$V(\mathcal{T}_{t,h_t}^{h_t}) \leq V(\mathcal{T}_{t,h}^h) + \mathcal{O}\left(\sqrt{\frac{K \ln \frac{T|\mathcal{F}|}{\delta}}{\epsilon h t}}\right).$$

On the other hand, from Lemma 9 and union bound over all time steps $t$ in $[T]$, we have that with probability $1 - \delta/2$,

$$V(\mathcal{T}_{t,h}^h) \leq \min_{\mathcal{T} \in \mathcal{F}_K} V(\mathcal{T}_K) + \mathcal{O}\left(\sqrt{\frac{K^2 \log K \cdot \left(\ln \frac{T|\mathcal{F}|}{\delta}\right)}{\epsilon h t}}\right).$$

Combining the above two inequalities, we have that with probability $1 - \delta$, for all $h$ in $\mathcal{H}$,

$$V(\mathcal{T}_{t,h_t}^{h_t}) \leq \min_{\mathcal{T} \in \mathcal{F}_K} V(\mathcal{T}_K) + \mathcal{O}\left(\sqrt{\frac{K^2 \log K \cdot \left(\ln \frac{T|\mathcal{F}|}{\delta}\right)}{\epsilon h t}}\right).$$

By the setting of $\mathcal{H}$, we can guarantee that the above also implies that the equation above holds for all $h \in (0, 1]$ (see [37, Lemma 20] for a detailed argument). By standard regret analysis of $\epsilon$-greedy exploration, this implies that for all $h \in (0, 1]$,

$$\mathrm{Reg}(T, \mathcal{F}_K, h) \leq \epsilon T + \mathcal{O}\left(\sum_{t=1}^{T} \sqrt{\frac{K^2 \log K \cdot \left(\ln \frac{T|\mathcal{F}|}{\delta}\right)}{\epsilon h t}}\right) = \mathcal{O}\left(\epsilon T + \sqrt{\frac{K^2 \log K \cdot T \cdot \left(\ln \frac{T|\mathcal{F}|}{\delta}\right)}{\epsilon h}}\right).$$

We conclude the first item, by the above inequality, and observing that for any tree policy in $\mathcal{F}_\infty$, there exists a tree policy in $\mathcal{F}_K$ that has extra $h$-smoothed expected loss at most $\frac{1}{hK}$.

The second item follows directly by the settings of $\epsilon$, $K$ and algebra. $\qquad \square$

## D  Algorithms for general policy classes

In this section, we generalize CATS and propose two algorithms, namely Algorithms 6 and 7, that works with general policy classes $\Pi$. On one hand, the two algorithms presented in this section may not be computationally efficient in general, because off-policy optimization w.r.t $\Pi$ can be computationally intractable; on the other hand, they have similar regret guarantees as CATS and Algorithm 5 while being able to handle policy classes beyond trees.

We first present Algorithm 6, an algorithm that naturally generalizes the $\epsilon$-greedy algorithm [e.g. 39] in the discrete action space setting to the continuous action space setting. It has two input parameters: a bandwidth parameter $h$, and a parameter $\epsilon \in [0, 1]$ that controls the exploration-exploitation tradeoff.

As we will see, given bandwidth parameter $h$, the algorithm provides a $h$-smoothed regret guarantee. Furthermore, if $\epsilon$ is large, the algorithm explores more, and learns more on the loss function at each round; in contrast, a choice of small $\epsilon$ lets the algorithm focuses more on exploitation, i.e. utilizing the learned policy more extensively.

The algorithm proceeds in rounds. At round $t$, it generates a stochastic policy $P_t$ that is a mixture of $\pi_{t,h}$ and the uniform distribution, where the mixture weights are $(1 - \epsilon)$ and $\epsilon$ respectively. Based

---

**Algorithm 6** Smoothed $\epsilon$-greedy algorithm with general policy classes

---

1: Input: Greedy parameter $\epsilon$, smoothing parameter $h$, policy class $\Pi$.
2: Let $\pi_1$ be an arbitrary policy in $\Pi$.
3: **for** $t = 1, 2, \ldots$ **do**
4:    Define policy $P_t(a|x) := (1 - \epsilon)\pi_{t,h}(a|x) + \epsilon$.
5:    Observe context $x_t$, select action $a_t \sim P_t(\cdot|x_t)$, observe loss $\ell_t(a_t)$.
6:    Find $\pi_{t+1} \leftarrow \operatorname{argmin}_{\pi \in \Pi} \hat{V}_t(\pi_h)$, where

$$\hat{V}_t(\pi_h) := \frac{1}{t} \sum_{s=1}^{t} \frac{\pi_h(a_s|x_s)}{P_s(a_s|x_s)} \ell_s(a_s).$$

---

on this policy, the algorithm selects an action $a_t \sim P_t(\cdot|x_t)$. After action $a_t$ is taken, the algorithm observes its loss incurred $\ell_t(a_t)$ and add the tuple $(x_t, a_t, P_t(a_t \mid x_t), \ell_t(a_t))$ into the interaction log. Then, it uses the interaction log collected up to round $t$ to build policy loss estimators $\hat{V}_t(\pi_h)$ for every policy $\pi$ in $\Pi$, which serves a proxy of $\pi_h$'s expected loss $\lambda_h(\pi)$. Then, it finds policy $\pi_{t+1}$ that minimizes $\hat{V}_t(\pi_h)$. The rationale is that, as $\hat{V}_t(\pi_h)$ concentrates around $\lambda_h(\pi)$ for all $\pi$, $\pi_{t+1}$ will also approximately minimize $\lambda_h(\cdot)$ among all policies in $\Pi$.

We have the following theorem that characterizes the $h$-smoothed regret of Algorithm 6.

**Theorem 13.** *Suppose Algorithm 6 is run with greedy parameter $\epsilon$, smoothing parameter $h$ and policy class $\Pi$. Then with probability $1 - \delta$, it has $h$-smoothed regret bounded as:*

$$\operatorname{Reg}(T, \Pi, h) \leq \tilde{\mathcal{O}}\left(\epsilon T + \sqrt{\frac{T}{\epsilon h} \cdot \left(\ln|\Pi| + \ln\frac{1}{\delta}\right)}\right).$$

*Furthermore, setting $\epsilon = \min\left(1, \left(\frac{\ln|\Pi| + \ln\frac{1}{\delta}}{hT}\right)^{\frac{1}{3}}\right)$, we have that*

$$\operatorname{Reg}(T, \Pi, h) \leq \tilde{\mathcal{O}}\left(\left(\frac{T^2}{h}\left(\ln|\Pi| + \ln\frac{1}{\delta}\right)\right)^{\frac{1}{3}} + \sqrt{\frac{T}{h} \cdot \left(\ln|\Pi| + \ln\frac{1}{\delta}\right)}\right).$$

The above theorem gives a regret bound or order $\left(\frac{T^2}{h}\ln|\Pi|\right)^{\frac{1}{3}}$, which is similar to the $\left(T^2 K \ln|\Pi|\right)^{\frac{1}{3}}$ regret bound by $\epsilon$-greedy algorithms obtained in the discrete $K$-action setting [See e.g. 39]. Intuitively, $\frac{1}{h}$ characterizes the difficulty of obtaining a $h$-smoothed regret guarantee, which serves as the counterpart of the action set size in the discrete action setting.

The most computationally expensive step of Algorithm 6 is line 6, where we find the policy $\pi$ in $\Pi$ that has the smallest IPS loss $V_t(\pi)$. As discussed in Section 2, if $\Pi$ consists of policies that takes actions in the discrete set $\{i/K\}_{i=0}^{K-1}$, the policy optimization problem can be cast as a CSMC problem, where heuristic algorithms that perform approximate ERM abound; indeed, the `Train_tree` procedure in `CATS` can be viewed as one such algorithm.

*Proof of Theorem 13.* We let $\pi_\star = \operatorname{argmin}_{\pi \in \Pi} V(\pi_h)$ denote the optimal policy in $\Pi$ after $h$-smoothing. In this notation, recall that the $h$-smoothed regret can be written as:

$$\operatorname{Reg}(\Pi, T, h) = \sum_{t=1}^{T} \left(\mathbb{E}\left[\ell_t(a_t)\right] - V(\pi_{\star,h})\right). \tag{18}$$

Define event

$$E = \left\{\text{for all } t \text{ in } [T] \text{ and all } \pi \text{ in } \Pi, \left|\hat{V}_t(\pi_h) - V(\pi_h)\right| \leq 8\sqrt{\frac{\ln\frac{2T|\Pi|}{\delta}}{t\epsilon h}} + 4\frac{\ln\frac{2T|\Pi|}{\delta}}{t\epsilon h}.\right\}$$

Using Lemma 16 with $\delta' = \frac{\delta}{T}$ for every $t = 1, 2, \ldots, T$, $p_{\min} = \epsilon$, along with union bound over all $t$'s in $[T]$, we get that $\mathbb{P}(E) \geq 1 - \delta$. We condition on event $E$ happening in the sequel. We first provide an excess loss bound for policy $\pi_{t,h}$. At time step $t + 1$, $\pi_{t+1,h}$ is an empirical risk minimizer, therefore:

$$\hat{V}_t(\pi_{t+1,h}) \leq \hat{V}_t(\pi_{\star,h}). \tag{19}$$

Hence,

$$V(\pi_{t+1,h}) \leq \hat{V}_t(\pi_{t+1,h}) + 8\sqrt{\frac{\ln\frac{2T|\Pi|}{\delta}}{t\epsilon h}} + 4\frac{\ln\frac{2T|\Pi|}{\delta}}{t\epsilon h}$$

$$\leq \hat{V}_t(\pi_{\star,h}) + 8\sqrt{\frac{\ln\frac{2T|\Pi|}{\delta}}{t\epsilon h}} + 4\frac{\ln\frac{2T|\Pi|}{\delta}}{t\epsilon h}$$

$$\leq V(\pi_{\star,h}) + 16\sqrt{\frac{\ln\frac{2T|\Pi|}{\delta}}{t\epsilon h}} + 8\frac{\ln\frac{2T|\Pi|}{\delta}}{t\epsilon h},$$

where the first inequality is from the definition of $E$, and $\pi_t \in \Pi$; the second inequality is from Equation (19); the third inequality is from the definition of $E$, and $\pi_\star \in \Pi$;

We now claim that

$$V(\pi_{t+1,h}) \leq V(\pi_{\star,h}) + 24\sqrt{\frac{\ln\frac{2T|\Pi|}{\delta}}{t\epsilon h}}. \tag{20}$$

This is from a standard case analysis, and the simple fact that $V(\pi_{t+1,h}) \leq 1$: if $\frac{\ln\frac{2T|\Pi|}{\delta}}{t\epsilon h} \geq 1$ the inequality is trivial; otherwise, $16\sqrt{\frac{\ln\frac{2T|\Pi|}{\delta}}{t\epsilon h}} + 8\frac{\ln\frac{2T|\Pi|}{\delta}}{t\epsilon h} \leq (16+8)\sqrt{\frac{\ln\frac{2T|\Pi|}{\delta}}{t\epsilon h}} = 24\sqrt{\frac{\ln\frac{2T|\Pi|}{\delta}}{t\epsilon h}}$.

We now conclude the regret bound. We first have the following upper bound on the algorithm's instantaneous loss $\mathbb{E}\ell_t(a_t)$:

$$\mathbb{E}[\ell_t(a_t)] = (1-\epsilon)\mathbb{E}_{(x_t,\ell_t)\sim D}\mathbb{E}_{a\sim\pi_{t,h}(\cdot|x_t)}[\ell_t(a)] + \epsilon\mathbb{E}_{(x_t,\ell_t)\sim D}\mathbb{E}_{a\sim U(\mathcal{A})}[\ell_t(a)]$$
$$\leq V(\pi_{t,h}) + \epsilon. \tag{21}$$

Combining Equations (18), (20), (21), along with algebra, we have:

$$\mathrm{Reg}(\Pi, T, h) \leq \sum_{t=1}^T \left(\epsilon + V(\pi_{t,h}) - V(\pi_{\star,h})\right)$$

$$\leq \epsilon T + 1 + \sum_{t=2}^T \left(24\sqrt{\frac{\ln\frac{2T|\Pi|}{\delta}}{(t-1)\epsilon h}}\right)$$

$$\leq \epsilon T + 1 + 48\sqrt{\frac{T\ln\frac{2T|\Pi|}{\delta}}{\epsilon h}}.$$

The theorem follows. $\qquad\square$

We next present Algorithm 7, which achieves $h$-smoothed regret guarantees against $\Pi$ for all $h$ in $(0, 1]$ *simultaneously*. It has the following key differences from Algorithm 6:

1. Instead of working with a fixed bandwidth $h$, it works with a set of bandwidths $\mathcal{H}$ that provides a covering of the set of bandwidths $(0, 1]$ we compete with.

2. Instead of finding a policy $\pi$ that minimizes $\hat{V}_t(\pi_h)$ for a fixed $h$, the algorithm first finds a minimizer of $\hat{V}_t(\pi_h)$ for every $h \in \mathcal{H}$ (namely $\pi_{t+1,h}$), and selects $\pi_{t+1}$ among the set $\{\pi_{t+1,h}\}_{h\in\mathcal{H}}$, using a structural risk minimization [55] procedure (line 9). Specifically, the choice of $h_{t+1}$ ensures that the expected loss of $\pi_{t+1,h_{t+1}}$ has competitive performance compared with those of the $\pi_h$'s, for all $\pi$ in $\Pi$ and all $h$ in $\mathcal{H}$. Here, the bandwidth-dependent penalty term $P(t, h) \triangleq 2\sqrt{\frac{\ln|\Pi|+\ln\frac{8T^4}{\delta}}{t\epsilon h}} + 3\frac{\ln|\Pi|+\ln\frac{8T^4}{\delta}}{t\epsilon h}$ is crucial, as it accounts for the different concentration rates from $\hat{V}_t(\pi_{t+1,h})$ to $V(\pi_{t+1,h})$ form different values of $h$.

**Theorem 14.** *Suppose Algorithm 7 is run with greedy parameter $\epsilon$ and policy class $\Pi$. Then with probability $1 - \delta$, the algorithm has smoothed regret guarantee simultaneously for all $h \in (0, 1]$:*

$$\mathrm{Reg}(T, \Pi, h) \leq \tilde{\mathcal{O}}\left( \epsilon T + \sqrt{\frac{T}{\epsilon h} \cdot \left( \ln |\Pi| + \ln \frac{1}{\delta} \right)} \right).$$

*Furthermore, setting $\epsilon = \min\left( 1, \left( \frac{\ln |\Pi| + \ln \frac{1}{\delta}}{T} \right)^{\frac{1}{3}} \right)$, we have that for all $h \in (0, 1]$:*

$$\mathrm{Reg}(T, \Pi, h) \quad \leq \quad \tilde{\mathcal{O}}\left( \frac{\left( T^2 \left( \ln |\Pi| + \ln \frac{1}{\delta} \right) \right)^{\frac{1}{3}}}{\sqrt{h}} \right).$$

Before proving the theorem, we make two important remarks:

1. Theorem 12 of [37] shows that a combination of `Corral` [4] with `EXP4` [6], using an appropriate tuning of learning rate, can obtain a uniform-$h$-smoothed regret of the same order, i.e. $\mathcal{O}\left( \frac{T^{2/3} \ln |\Pi|^{\frac{1}{3}}}{\sqrt{h}} \right)$. However, their algorithm requires explicit enumeration of policies from policy class $\Pi$; in contrast, our algorithm can be reduced to a sequence of policy optimization problems, which can admit much more efficient implementations.

2. The above uniform-$h$-smoothed regret rate in terms of $h$ and $T$, i.e. $\mathcal{O}\left( \frac{T^{2/3}}{\sqrt{h}} \right)$, is unimprovable in general, and is therefore *Pareto optimal*. This can be seen from the following result from [37, Theorem 11]: there exists a continuous-action CB problem with action space $[0, 1]$, constants $c, T_0 > 0$, such that for any algorithm and any $T \geq T_0$, there exist two bandwidths $h_1 = \Theta(1)$ and $h_2 = o(1)$[10] such that $\mathrm{Reg}(T, \Pi, h_1) > \frac{cT^{2/3}}{\sqrt{h_1}}$ or $\mathrm{Reg}(T, \Pi, h_2) > \frac{cT^{2/3}}{\sqrt{h_2}}$. As a result, for any $\alpha > 0$, designing an algorithm that obtains a uniform-$h$-smoothed-regret guarantee of order $\mathcal{O}\left( \frac{T^{\frac{2}{3} - \alpha}}{h^{\frac{1}{2}}} \right)$ or order $\mathcal{O}\left( \frac{T^{\frac{2}{3}}}{h^{\frac{1}{2} - \alpha}} \right)$ is impossible. This result is perhaps surprising, as it shows that an $\epsilon$-greedy algorithm, well known to have suboptimal regret guarantees in the discrete action CB setting, possesses certain optimality properties in the continuous action CB setting, with appropriate modifications.

*Proof sketch.* By standard analysis on structural risk minimization [see e.g. 55], it can be shown that with high probability, for all $h \in \mathcal{H}$:

$$V(\pi_{t+1, h_{t+1}}) \leq \min_{\pi \in \Pi} V(\pi_h) + \mathcal{O}\left( \sqrt{\frac{\ln \frac{T |\Pi|}{\delta}}{t \epsilon h}} \right).$$

By the setting of $\mathcal{H} = \left\{ h \in \left\{ \frac{1}{4T^2}, \frac{2}{4T^2}, \ldots, 1 \right\} : h \geq \frac{1}{2T} \right\}$, we can show that that the above guarantee implies that the equation above holds for all $h \in (0, 1]$; see [37, Lemma 20] for a detailed proof.

By standard regret analysis of $\epsilon$-greedy algorithms and the above upper bound on the instantenous loss of $\pi_{t+1, h_{t+1}}$, we get that

$$\mathrm{Reg}(T, \Pi, h) \leq \epsilon T + \mathcal{O}\left( \sum_{t=1}^{T} \sqrt{\frac{\ln \frac{T |\Pi|}{\delta}}{t \epsilon h}} \right) = \mathcal{O}\left( \epsilon T + \sqrt{T \frac{\ln \frac{T |\Pi|}{\delta}}{\epsilon h}} \right).$$

The second item follows directly by the setting of $\epsilon$ and algebra. $\square$

**Algorithm 7** A Pareto-optimal adaptive-$h$ algorithm

---

1: Input: Greedy parameter $\epsilon$, policy class $\Pi$.
2: Let $\mathcal{H} = \left\{ h \in \left\{ \frac{1}{4T^2}, \frac{2}{4T^2}, \ldots, 1 \right\} : h \geq \frac{1}{2T} \right\}$ be the set of bandwidths in consideration.
3: Let $\pi_1$ be an arbitrary policy in $\Pi$, and $h_1$ be an arbitrary number in $\mathcal{H}$.
4: **for** $t = 1, 2, \ldots, T$ **do**
5:    Define policy $P_t(a|x) := (1 - \epsilon)\pi_{t,h_t}(a|x) + \epsilon$.
6:    Observe context $x_t$, select action $a_t \sim P_t(\cdot|x_t)$, observe loss $\ell_t(a_t)$.
7:    For every $h$ in $\mathcal{H}$, compute $\pi_{t+1}^h \in \Pi$ such that

$$\hat{V}_t(\pi_{t+1,h}^h) \leq \min_{\pi \in \Pi} \hat{V}_t(\pi_{t,h}^h), \tag{22}$$

   where

$$\hat{V}_t(\pi_h) \triangleq \frac{1}{t} \sum_{s=1}^{t} \frac{\pi_h(a_s|x_s)}{P_s(a_s|x_s)} \ell_s(a_s).$$

8:    Select $\pi_{t+1} = \pi_{t+1}^{h_{t+1}}$, where

$$h_t \in \operatorname*{argmin}_{h \in \mathcal{H}} \left( \hat{V}_t(\pi_{t+1,h}^h) + P(t,h) \right).$$

   where

$$P(t,h) \triangleq 2\sqrt{\frac{\ln|\Pi| + \ln \frac{8T^4}{\delta}}{t\epsilon h}} + 3\frac{\ln|\Pi| + \ln \frac{8T^4}{\delta}}{t\epsilon h}.$$

---

# E    Concentration inequalities

We first recall a well-known variant of Freedman's inequality [24, 8] that is useful to establish our policy evaluation concentration bounds.

**Lemma 15** (See [8], Lemma 2). *Suppose $X_1, \ldots, X_n$ is a martingale difference sequence adapted to filtration $\{\mathcal{B}_i\}_{i=0}^n$, where $|X_i| \leq M$ almost surely. Denote by $V_n = \sum_{j=1}^n \mathbb{E}\left[ X_j^2 \mid \mathcal{B}_{j-1} \right]$. Then for any constant $\delta \in (0, \frac{1}{e})$, with probability $1 - \delta$,*

$$\left| \sum_{i=1}^n X_i \right| \leq 4\sqrt{V_n \ln \frac{2n}{\delta}} + 2M \ln \frac{2n}{\delta}. \tag{23}$$

*Proof.* Lemma 2 of [8] states that for any $\delta' \in (0, \frac{1}{e})$, with probability $1 - \delta' \cdot \log n$,

$$\sum_{i=1}^n X_i \leq \max\left( 4\sqrt{V_n \ln \frac{1}{\delta'}}, 2\ln \frac{1}{\delta'} \right)$$

Letting $\delta' = \frac{\delta}{2\log n}$, we have that with probability $1 - \delta/2$,

$$\sum_{i=1}^n X_i \leq \max\left( 4\sqrt{V_n \ln \frac{1}{\delta'}}, 2\ln \frac{1}{\delta'} \right) \leq 4\sqrt{V_n \ln \frac{2n}{\delta}} + 2M \ln \frac{2n}{\delta},$$

where the second inequality is by algebra and the fact that $\log n \leq n$. Similarly, by considering random variable $\{-X_i\}_{i=1}^n$, we have that with probability $1 - \delta/2$,

$$\sum_{i=1}^n X_i \geq -\left( 4\sqrt{V_n \ln \frac{2n}{\delta}} + 2M \ln \frac{2n}{\delta} \right),$$

The lemma is concluded by union bound. $\qquad \square$

The above lemma implies the following important concentration result on off-policy evaluation and optimization. First we set up some notations.

Suppose logged data $\{(x_s, a_s, P_s(a_s \mid x_s), \ell_s(a_s)) : s \in [t]\}$ is $p_{\min}$-well-formed (recall Definition 8). Define a filtration $\{\mathcal{B}_s\}_{s=0}^t$ as follows: for all $s \in \{0, 1, \ldots, t\}$, $\mathcal{B}_s \triangleq \sigma(x_1, a_1, \ell_1, \ldots, x_s, a_s, \ell_s)$. A sequence of random variables $\{Z_s\}_{s=1}^t$ is said to be *predictable* w.r.t. filtration $\{\mathcal{B}_s\}_{s=1}^t$ if $Z_s$ is $\mathcal{B}_{s-1}$-measurable. Using the above notation, we see that the sequence of logging policies $\{P_s\}_{s=1}^t$ is predictable wrt $\{\mathcal{B}_s\}_{s=0}^t$. Lastly, recall from Section 2 that $\tilde{c}_s^h(i/K) = \frac{\text{Smooth}_h(a_s \mid i/K)}{P_s(a_s \mid x_s)} \ell_s(a_s)$ for $i \in \{0, 1, \ldots, K-1\}$, and therefore, $\tilde{c}_s^h(\pi(x_s)) = \frac{\pi_h(a_s \mid x_s)}{P_s(a_s \mid x_s)} \ell_s(a_s)$.

**Lemma 16.** *Suppose the setting is described as above. Then,*

1. *With probability $1 - \delta'$, we have that for any sequence of policies $\{\pi_s\}_{s=1}^t$ predictable w.r.t. $\{\mathcal{B}_s\}_{s=0}^t$,*

$$\left| \frac{1}{t} \sum_{s=1}^t \tilde{c}_s^h(\pi_s(x_s)) - \frac{1}{t} \sum_{s=1}^t V(\pi_{s,h}) \right| \leq \sqrt{\left( \frac{1}{t} \sum_{s=1}^t V(\pi_{s,h}) \right) \frac{16 \ln \frac{2t}{\delta'}}{t \, p_{\min} \, h}} + \frac{2 \ln \frac{2t}{\delta'}}{t \, p_{\min} \, h}. \quad (24)$$

2. *Given a finite set of policies $\Pi$, with probability $1 - \delta'$, for all $\pi$ in $\Pi$,*

$$\left| \hat{V}_t(\pi_h) - V(\pi_h) \right| \leq 4 \sqrt{\frac{\ln |\Pi| + \ln \frac{2t}{\delta'}}{t \, p_{\min} \, h}} + 2 \frac{\ln |\Pi| + \ln \frac{2t}{\delta'}}{t \, p_{\min} \, h}. \quad (25)$$

*Proof.* For the first item, we define $X_s \triangleq \frac{\pi_{s,h}(a_s \mid x_s)}{P_s(a_s \mid x_s)} \ell_s(a_s)$. In this notation, $\frac{1}{t} \sum_{s=1}^t \tilde{c}_s^h(\pi_s(x_s)) = \frac{1}{t} \sum_{s=1}^t X_s$. Observe that

$$\mathbb{E}[X_s \mid \mathcal{B}_{s-1}] = \mathbb{E}_{(x_s, \ell_s) \sim \mathcal{D}} \mathbb{E}_{a_s \sim P_s(\cdot \mid x_s)} \frac{\pi_{s,h}(a_s \mid x_s)}{P_s(a_s \mid x_s)} \ell_s(a_s)$$

$$= \mathbb{E}_{(x_s, \ell_s) \sim \mathcal{D}} \mathbb{E}_{a_s \sim \pi_{s,h}(\cdot \mid x_s)} \ell_s(a_s) = V(\pi_{s,h}). \quad (26)$$

Let $Z_s = X_s - \mathbb{E}[X_s \mid \mathcal{B}_{s-1}] = X_s - V(\pi_{s,h})$. It can be seen that $\{Z_s\}_{s=1}^t$ is a martingale difference sequence adapted to filtration $\{\mathcal{B}_s\}_{s=0}^t$.

Let $M = \frac{1}{h \, p_{\min}}$; From the definition of $Z_s$, along with the facts that $P_s(a_s \mid x_s) \geq p_{\min}$, and $\pi_{s,h}(a_s \mid x_s) \in [0, \frac{1}{h}]$ with probability 1, we get that $|Z_s(\pi)| \leq M$ with probability 1.

We now show an upper bound on the conditional variance of $Z_s$:

$$\mathbb{E}\left[Z_s^2 \mid \mathcal{B}_{s-1}\right] \leq \mathbb{E}\left[X_s^2 \mid \mathcal{B}_{s-1}\right]$$

$$= \mathbb{E}_{(x_s, \ell_s) \sim D} \mathbb{E}_{a_s \sim P_s(\cdot \mid x_s)} \left[ \frac{\pi_{s,h}(a_s \mid x_s)^2}{P_s(a_s \mid x_s)^2} \ell_s(a_s)^2 \right]$$

$$\leq \mathbb{E}_{(x_s, \ell_s) \sim D} \mathbb{E}_{a_s \sim P_s(\cdot \mid x_s)} \left[ \frac{\pi_{s,h}(a_s \mid x_s)^2}{P_s(a_s \mid x_s)^2} \ell_s(a_s) \right]$$

$$= \mathbb{E}_{(x_s, \ell_s) \sim D} \left[ \int_{[0,1]} \frac{\pi_{s,h}(a \mid x_s)^2}{P_s(a \mid x_s)^2} P_s(a \mid x_s) \ell_s(a) \, \mathrm{d}a \right]$$

$$= \mathbb{E}_{(x_s, \ell_s) \sim D} \int_{[0,1]} \frac{\pi_{s,h}(a \mid x_s)}{P_s(a \mid x_s)} \pi_{s,h}(a \mid x_s) \ell_s(a) \, \mathrm{d}a$$

$$\leq \mathbb{E}_{(x_s, \ell_s) \sim D} \frac{1}{p_{\min} h} \cdot \int_{[0,1]} \pi_{s,h}(a \mid x_s) \ell_s(a) \, \mathrm{d}a = \frac{V(\pi_{s,h})}{p_{\min} h}.$$

where the first inequality uses the fact that $\ell_s(a_s) \in [0, 1]$, and the second inequality uses the facts that $\pi_h(a \mid x_s) \in [0, \frac{1}{h}]$, and $P_s(a_s \mid x_s) \geq p_{\min}$. Consequently, $\sum_{s=1}^t \mathbb{E}\left[Z_s^2 \mid \mathcal{B}_{s-1}\right] \leq \frac{1}{p_{\min} h} \sum_{s=1}^t V(\pi_{s,h})$.

Applying Lemma 15 on $Z_s$'s, with $n = t$, $M = \frac{1}{hp_{\min}}$, $\delta = \delta'$, we have that with probability $1 - \delta'$:

$$\left| \sum_{s=1}^{t} \tilde{c}_s^h(\pi_s(x_s)) - \sum_{s=1}^{t} V(\pi_{s,h}) \right| \leq 4\sqrt{\left( \sum_{s=1}^{t} V(\pi_{s,h}) \right) \frac{\ln \frac{2t}{\delta'}}{p_{\min} h}} + \frac{2 \ln \frac{2t}{\delta'}}{p_{\min} h}.$$

The first item now follows from dividing both sides of the above inequality by $t$.

We now use the first item to show the second item. Fix a $\pi$ in $\Pi$. We take $\{\pi_s\}_{s=1}^{t}$ such that $\pi_s = \pi$ for all $s$. By the previous item, we have that with probability $1 - \frac{\delta'}{|\Pi|}$,

$$\left| \frac{1}{t} \sum_{s=1}^{t} \tilde{c}_s^h(\pi(x_s)) - \frac{1}{t} \sum_{s=1}^{t} V(\pi_h) \right| \leq 4\sqrt{\left( \frac{1}{t} \sum_{s=1}^{t} V(\pi_h) \right) \frac{\ln \frac{2|\Pi|t}{\delta'}}{t \, p_{\min} \, h}} + \frac{2 \ln \frac{2|\Pi|t}{\delta'}}{t \, p_{\min} \, h}$$

$$\leq 4\sqrt{\frac{\ln \frac{2|\Pi|t}{\delta'}}{t \, p_{\min} \, h}} + \frac{2 \ln \frac{2|\Pi|t}{\delta'}}{t \, p_{\min} \, h}.$$

We conclude the item by taking a union bound on all $\pi$ in $\Pi$. $\qquad\square$

# F   Experimental Details

Of the six datasets five were selected randomly from OpenML with the criterion of having millions of samples with unique regression values. These include `wisconsin`, `cpu_act`, `auto_price`, `black_friday` (customer purchases on black Friday) and `zurich_delay` (Zurich public transport delay data). We also included a synthetic dataset, namely `ds`, which was created by linear regression of standard gaussians with additive noise.

Our main comparator is the discretized $\epsilon$-greedy algorithm `dLinear` in Vowpal Wabbit which by default uses the doubly robust approach [22] for policy evaluation and optimization. This method reduces to cost-sensitive one-against-all multi-class classification which has computational complexity linear w.r.t number of discrete actions. Our other comparator is `dTree`, the discretized filter tree which is equivalent to `CATS` without smoothing, i.e. with zero bandwidth. For all the approaches we used $\epsilon = 0.05$ and a parameter free update rule based on coin betting [46].

We implemented `CATS` in Vowpal Wabbit. The details of the implementation are explained in the next section.

# G   CATS implementation with $\mathcal{O}(\log K)$ time per example

In this section, we present the details of our online implementation of `CATS` that has $\mathcal{O}(\log K)$ time cost per example. Our implementation can be generalized to the setting where the action space $\mathcal{A}$ is a continuous interval in $\mathbb{R}$; for simplicity of presentation, we focus on $\mathcal{A} = [0, 1]$ in this section. Before going into the details, we introduce some additional notation.

Recall that $K = 2^D$ is the discretization level; the corresponding discretized action space is defined as $\mathcal{A}_K = \left\{ 0, \frac{1}{K}, \ldots, \frac{K-1}{K} \right\}$. We will consider choices of bandwidth $h$ in $\mathcal{H}_K = \left\{ 2^{-i} : i \in [K] \right\}$; our algorithm can be easily generalized to other values of $h$'s, by modifying the tree initialization procedure. For a bandwidth $h$ in $\mathcal{H}_K$, define an auxiliary parameter $m^{\#} = \log_2(K \cdot h)$, which is an integer. It can be easily seen that $h = 2^{m^{\#}}/K$.

## G.1   `Build_tree`: initialization of tree policy

We now describe a procedure `Build_tree`, namely Algorithm 9, that provides essential initialization of our tree policy $\mathcal{T}$. First, `Build_tree` assigns a unique `id` for each node v in the tree $\mathcal{T}$ through traversing the tree in a top-down fashion. It also supplies the action labels of all $K$ leaves. The nodes' `id`'s are assigned such that within the same level, the `id`'s are increasing from left to right. Furthermore, it initializes the online binary CSMC base learners in all its internal nodes. To ensure $\mathcal{O}(\log K)$ time cost of the tree learning algorithm, we disallow actions in $\mathcal{A}_K \cap [0, h]$ and $\mathcal{A}_K \cap [1 - h, 1]$ to be taken by the tree policy. To this end, two classifiers in nodes $\text{v}^{\text{only\_left}}$ and $\text{v}^{\text{only\_right}}$ are set to fixed classifiers $f^{\text{v}^{\text{only\_left}}} \equiv \texttt{left}$ and $f^{\text{v}^{\text{only\_right}}} \equiv \texttt{right}$, both of which are *read-only*.

---

**Algorithm 8** `Train_tree` with no data partitioning

---

1: Input: $K = 2^D$, $\mathcal{F}$, training data $\{(x_s, c_s)\}_{s=1}^n$ with $c_s \in \mathbb{R}^K$
2: **for** level $d$ from $D-1$ down to 0 **do**
3:     **for** nodes v at level $d$ **do**
4:         For each $(x_s, c_s)$ define binary cost $c_s^{\text{v}}$ with

$$c_s^{\text{v}}(\texttt{left}) = c_s(\texttt{v.left.get\_action}(x_s))$$
$$c_s^{\text{v}}(\texttt{right}) = c_s(\texttt{v.right.get\_action}(x_s)).$$

5:         Train $f^{\text{v}} \in \mathcal{F}$ on $S^{\text{v}} = \{(x_s, c_s^{\text{v}}) : s \in [n], c_s^{\text{v}}(\texttt{left}) \neq c_s^{\text{v}}(\texttt{right})\}$:

$$f^{\text{v}} \in \underset{f \in \mathcal{F}}{\operatorname{argmin}}\, \mathbb{E}_{S^{\text{v}}}\left[c^{\text{v}}(f(x))\right].$$

6: Return tree $\mathcal{T}$ with $\{f^{\text{v}}\}$ as node classifiers.

---

---

**Algorithm 9** `Build_tree`

---

**Input:** Tree $\mathcal{T}$ with depth $D$, $m^{\#}$ {Initialize a tree policy $\mathcal{T}$ with $K = 2^D$ leaves by assigning id's to each node; in addition, initialize the nodes such that the leftmost and rightmost $2^{m^{\#}}$ leaves are unreachable}
**Output:** Initialized tree policy $\mathcal{T}$
  $\mathcal{T}$.root.id $\leftarrow 0$
  **for** level $d$ in $\{0, \ldots, D-1\}$ **do**
    **for** nodes v at level $d$ of $\mathcal{T}$ **do**
      Initialize the online CSMC base learner at v
      v.left.id $= 2 \times$ v.id $+ 1$
      v.right.id $= 2 \times$ v.id $+ 2$
  **for** nodes v at level $D$ of $\mathcal{T}$ **do**
    Set label(v) $\leftarrow$ (v.id $- (2^D - 1))/K$.
  Set $\text{v}^{\texttt{only\_right}}$ to be the node v with v.id $= 2^{D-m^{\#}-1} - 1$, and let $f^{\text{v}^{\texttt{only\_right}}} \equiv \texttt{right}$.
  Set $\text{v}^{\texttt{only-left}}$ to be the node v with v.id $= 2^{D-m^{\#}} - 2$, and let $f^{\text{v}^{\texttt{only.left}}} \equiv \texttt{left}$.

---

To see why the above restriction helps with ensuring $\mathcal{O}(\log K)$ time cost per example, we now recall the definition of the IPS CSMC example $(x_t, \tilde{c}_t)$ generated by log data $(x_t, a_t, \ell_t(a_t), P_t(a_t \mid x_t))$ in `CATS`. We first show that $\tilde{c}_t$ has a simple structure: if $a$ is in $\mathcal{A}_K \cap [h, 1-h]$, we have a concise formula of $\tilde{c}_t(a)$:

$$\tilde{c}_t(a) = \begin{cases} \frac{\ell_t(a_t)}{2hP_t(a_t|x_t)}, & |a - a_t| \leq h, \\ 0, & \text{otherwise.} \end{cases} \tag{27}$$

Observe that $\tilde{c}_t$ is a piecewise constant function over $\mathcal{A}_K \cap [h, 1-h]$ with at most 3 pieces: $[0, a_t - h]$ (if $a_t > h$), $[\max(0, a_t - h), \min(1, a_t + h)]$, and $[a_t + h, 1]$ (if $a_t < 1 - h$). The IPS cost vector $\tilde{c}_t$ can be summarized by three numbers: $c^* = \frac{\ell_t(a_t)}{2hP_t(a_t|x_t)}$, the nonzero value in $\tilde{c}_t$, $a_{\min} = \max(0, \frac{\lceil K(a_t - h) \rceil}{K})$, the minimum $a \in \mathcal{A}_K$ such that $\tilde{c}_t(a) = c^*$; $a_{\max} = \min(\frac{K-1}{K}, \frac{\lfloor K(a_t + h) \rfloor}{K})$, the maximum $a \in \mathcal{A}_K$ such that $\tilde{c}_t(a) = c^*$.

We remark that $\tilde{c}_t$ may not be a piecewise constant function *globally* over $\mathcal{A}_K$. This is because in general, $\tilde{c}_t(a) = \frac{\ell_t(a_t)\texttt{Smooth}(a_t|a)}{P_t(a_t|x_t)} = \frac{\ell_t(a_t)\mathbf{1}(a - a_t \leq h)}{\text{vol}([a-h, a+h] \cap [0,1]) \cdot P_t(a_t|x_t)}$, where $\text{vol}(\cdot)$ denotes the Lebesgue measure. Therefore, if, say $a_t$ is in $[0, h]$, the induced IPS cost function $\tilde{c}_t$ can take many possible positive values for $a$ in region $[0, h]$, depending on the value of $\text{vol}([a-h, a+h] \cap [0,1])$. It turns out that enforcing the piecewise constant structure of the cost vector (as is done by restricting the CSMC vectors to only consider entries in $a$ in $\mathcal{A}_K \cap [h, 1-h]$) is vital to achieve $\mathcal{O}(\log K)$ per-example time cost, as we will see next.

---

**Algorithm 10** `Online_train_tree` with $\mathcal{O}(\log K)$ time cost per example

---

**Input:** Tree policy $\mathcal{T}$, context $x$, cost vector $\tilde{c}$ implicitly represented by actions $a_{\min}, a_{\max}$ in $\mathcal{A}_K$ and cost $c^*$ in $\mathbb{R}_+$, such that for all $a \in \mathcal{A}_K$, $\tilde{c}(a) = c^*$ if $a \in [a_{\min}, a_{\max}]$, and $\tilde{c}(a) = 0$ otherwise.

**Output:** Updated tree policy $\mathcal{T}$.

1: $\alpha \leftarrow$ leaf corresponding to action $a_{\min}$, $\beta \leftarrow$ leaf corresponding to action $a_{\max}$
2: $\alpha.\texttt{cost} \leftarrow c^*$, $\beta.\texttt{cost} \leftarrow c^*$
3: $\alpha_D \leftarrow \alpha$, $\beta_D \leftarrow \beta$.
4: **for** level $d$ from $D$ down to 1 **do**
5:    **if** $\alpha_d.\texttt{parent} \neq \beta_d.\texttt{parent}$ **then**
6:       $S_d \leftarrow \{\alpha_d, \beta_d\}$
7:    **else**
8:       $S_d \leftarrow \{\alpha_d\}$;
9:    **for** nodes $\texttt{v} \in S_d$ **do**
10:       $\texttt{u} \leftarrow \texttt{v.parent}$ {Goal: update the online learner in $\texttt{u}$, the parent of $\texttt{v}$}
11:       **if** $\texttt{u} \in \{\texttt{v}^{\texttt{only\_left}}, \texttt{v}^{\texttt{only\_right}}\}$ **then**
12:          **continue**; {No updates on $\texttt{v}^{\texttt{only\_left}}$ and $\texttt{v}^{\texttt{only\_right}}$}
13:       $\texttt{w} \leftarrow$ the sibling of node $\texttt{v}$. {Create cost vector $c^{\texttt{u}}$}
14:       $\texttt{w.cost} \leftarrow \texttt{Return\_cost}(\texttt{w}, \alpha_d, \beta_d)$
15:       **if** $\texttt{v} = \texttt{u.left}$ **then**
16:          $c^{\texttt{u}}(\texttt{left}) \leftarrow \texttt{v.cost}, c^{\texttt{u}}(\texttt{right}) \leftarrow \texttt{w.cost}$ {$\texttt{v}$ is the left child of $\texttt{u}$}
17:       **else**
18:          $c^{\texttt{u}}(\texttt{left}) \leftarrow \texttt{w.cost}, c^{\texttt{u}}(\texttt{right}) \leftarrow \texttt{v.cost}$ {$\texttt{v}$ is the right child of $\texttt{u}$}
19:       $\texttt{u.learn}(f^{\texttt{u}}, (x, c^{\texttt{u}}))$ {Update the online CSMC base learner in $\texttt{u}$}
20:       $\texttt{u.cost} \leftarrow c^{\texttt{u}}(f^{\texttt{u}}(x))$ {Compute $c(\mathcal{T}^{\texttt{u}}(x))$ for training in nodes of higher level}
21:    $\alpha_{d-1} \leftarrow \alpha_d.\texttt{parent}$ {Compute the ancestors of $\alpha, \beta$ to a level up}
22:    $\beta_{d-1} \leftarrow \beta_d.\texttt{parent}$

---

---

**Algorithm 11** `Return_cost`

---

**Input:** Tree node $\texttt{w}$; Tree nodes $\alpha$ and $\beta$, which are the ancestors of $\alpha$ and $\beta$, respectively, at the same level of $\texttt{w}$.

   **if** $\texttt{w.id} < \alpha.\texttt{id}$ or $\texttt{w.id} > \beta.\texttt{id}$ **then**
      **return** $0$
   **else if** $\alpha.\texttt{id} < \texttt{w.id} < \beta.\texttt{id}$ **then**
      **return** $c^*$
   **else if** $\texttt{w.id} = \alpha.\texttt{id}$ **then**
      **return** $\alpha.\texttt{cost}$
   **else if** $\texttt{w.id} = \beta.\texttt{id}$ **then**
      **return** $\beta.\texttt{cost}$

---

### G.2 `Online_train_tree`: online update of tree policy

In our implementation, to maximize data-efficiency, we will implement a more practical variant of `Train_tree`, namely Algorithm 8; the difference between it and `Train_tree` is that, instead of partitioning the input data to train each level separately, we use the full input data to train nodes at all levels.

The tree policy training algorithm, namely `Online_train_tree` (Algorithm 10), is an online implementation of Algorithm 8. It is used by CATS (in its line 7) to process the IPS CSMC example generated at every round $t$, to obtain an updated tree policy. It receives a IPS CSMC example $(x, \tilde{c})$ as input, represented by context $x$, and $a_{\min}, a_{\max}, c^*$ (representing $\tilde{c}$, as discussed in the previous section), and a tree $\mathcal{T}$ trained over previous CSMC examples $S$; specifically, $(x_t, \tilde{c}_t)$'s in CATS are its valid inputs. Here we assume that the input $\mathcal{T}$ is such that for every node $\texttt{v}$, its stored classifier $f^{\texttt{v}}$ is an approximation of $\operatorname{argmin}_{f \in \mathcal{F}} \mathbb{E}_{(x,c) \sim S}[c^{\texttt{v}}(f(x))]$ (recall the definition of $c^{\texttt{v}}$ in Algorithm 8). `Online_train_tree` updates the input $\mathcal{T}$ with $(x, \tilde{c})$, such that it approximates the output of `Train_tree` over $S \cup \{(x, \tilde{c})\}$, that is, for every node $\texttt{v}$, its stored classifier $f^{\texttt{v}}$ is an approximation of $\operatorname{argmin}_{f \in \mathcal{F}} \mathbb{E}_{(x,c) \sim S \cup \{(x,\tilde{c})\}}[c^{\texttt{v}}(f(x))]$. Our online implementation replaces line 7

of CATS with $\mathcal{T} \leftarrow \texttt{Online\_train\_tree}(\mathcal{T}, (x_t, \tilde{c}_t))$, with the goal of ensuring the updated $\mathcal{T}$ after round $t$ closely approximates $\texttt{Train\_tree}(K, \mathcal{F}, \{(x_s, \tilde{c}_s)\}_{s=1}^t)$.

The tree policy update proceeds in a bottom-up fashion. Given two leaves of the tree $\alpha, \beta$ that correspond to actions $a_{\min}, a_{\max}$, we use them as "seeds" to "climb up" the tree, reaching nodes that need updating. Specifically, for every level $d \in [D]$, we maintain $\alpha_d$ and $\beta_d$ that correspond to the ancestors of $\alpha$ and $\beta$, respectively, at that level.

As discussed in the main text, for a given node v, if $c^{\mathtt{v}}(\texttt{left}) = c^{\mathtt{v}}(\texttt{right})$, there is no need to update the online CSMC learner at v, because $f_{t+1}^{\mathtt{v}}$, the ERM at node v at time $t+1$, will be equal to $f_t^{\mathtt{v}}$. From Lemma 17 below, it turns out that it suffices to only update the CSMC online learners in $\alpha_d$'s and $\beta_d$'s at levels $d \in \{0, \dots, D-1\}$. In addition, to update an internal node v, one needs to obtain $c^{\mathtt{v}}(\texttt{left})$ and $c^{\mathtt{v}}(\texttt{right})$, which corresponds to costs of the action routed by its left and right subtrees, i.e. $\tilde{c}(\mathcal{T}^{\mathtt{v.left}}(x))$ and $\tilde{c}(\mathcal{T}^{\mathtt{v.right}}(x))$. To ensure computational efficiency, Algorithm 10 calls a carefully-designed subprocedure, namely $\texttt{Return\_cost}$ (Algorithm 11), that given any node v at level $d$, returns the cost $\tilde{c}(\mathcal{T}^{\mathtt{v}}(x))$ in *constant time*, provided that $\alpha_d, \beta_d$, the ancestors of $\alpha, \beta$ at the level $d$, have been identified. We refer the reader to Claim 18 for a proof of correctness of $\texttt{Return\_cost}$. Upon receiving binary CSMC example $(x, c^{\mathtt{v}})$, the CSMC oracle at node v gets updated using an incremental update rule (such as stochastic gradient descent) on $(x, c^{\mathtt{v}})$ at line 19 of $\texttt{Online\_train\_tree}$, which we assume takes $\mathcal{O}(1)$ time (where the $\mathcal{O}(\cdot)$ notation here is only with respect to the discretization level $K$). Specifically, our implementation of CATS in Vowpal Wabbit uses base CSMC learners that performs a reduction from classification to online least-squares regression to approximate ERM: at every node, its corresponding base learner learns to predict the cost of going to the left and right branch respectively, and the learned classifier takes the branch with lower predicted cost. Furthermore, we use a parameter-free gradient update rule [46] to implement our online least square regression procedure. As a result, in our implementation, the time costs of each base learner's prediction and update are both $\mathcal{O}(d)$, where $d$ is dimension of the context space.

We finally remark that in line 12 of Algorithm 10, we skip updates on nodes $\mathtt{v}^{\mathtt{only\_left}}$ and $\mathtt{v}^{\mathtt{only\_right}}$, ensuring that the tree policy never outputs actions in $\mathcal{A}_K \cap [0, h]$ or $\mathcal{A}_K \cap [1-h, 1]$.

### G.2.1 Proof of correctness of $\texttt{Online\_train\_tree}$

We now prove that Algorithm 10 does not miss updating nodes that needs updates, i.e. the nodes u such that $c^{\mathtt{u}}(\texttt{left}) \neq c^{\mathtt{u}}(\texttt{right})$; recall that $c^{\mathtt{u}}(\texttt{left}) = \tilde{c}(\mathcal{T}^{\mathtt{u.left}}(x))$ and $c^{\mathtt{u}}(\texttt{right}) = \tilde{c}(\mathcal{T}^{\mathtt{u.right}}(x))$.

**Lemma 17.** *For every internal node u in $\mathcal{T}$, if $c^{\mathtt{u}}(\texttt{left}) \neq c^{\mathtt{u}}(\texttt{right})$, then Algorithm 10 updates* u *with binary cost-sensitive example $(x, c^{\mathtt{u}})$. Consequently,* $\texttt{Online\_train\_tree}$ *(Algorithm 10) faithfully implements* $\texttt{Train\_tree}$ *(Algorithm 2) in an online fashion.*

*Proof.* With the notations defined in $\texttt{Online\_train\_tree}$ (Algorithm 10), denote by $\alpha$ (resp. $\beta$) the leaf with action label $a_{\min}$ (resp. $a_{\max}$). It can be seen from the description of $\texttt{Online\_train\_tree}$ that if node u is an ancestor of $\alpha$ or $\beta$, the base CSMC learner in u will get updated. We now show that if $c^{\mathtt{u}}(\texttt{left}) \neq c^{\mathtt{u}}(\texttt{right})$, u must be an ancestor of either $\alpha$ or $\beta$, which will let us conclude that all nodes u with $c^{\mathtt{u}}(\texttt{left}) \neq c^{\mathtt{u}}(\texttt{right})$ will be updated.

We will prove the above statement's contrapositive: if neither $\alpha$ nor $\beta$ is a child of u, then $c^{\mathtt{u}}(\texttt{left}) = c^{\mathtt{u}}(\texttt{right})$. Indeed, suppose u is at level $d$, and denote by $\alpha_d$ and $\beta_d$ the ancestors of $\alpha, \beta$ at level $d$ respectively. Then, it must be the case that $\mathtt{u} \neq \alpha_d$ and $\mathtt{u} \neq \beta_d$. From the first two items of Claim 18 below, we have that $\tilde{c}(a)$ must agree unanimously for all actions $a$ in $\text{range}(\mathcal{T}^{\mathtt{u}})$. Now, because both $c^{\mathtt{u}}(\texttt{left})$ and $c^{\mathtt{u}}(\texttt{right})$ take values in $\text{range}(\mathcal{T}^{\mathtt{u}})$, they must also be equal.

In addition, from the last item in Claim 18 below, along with the description of $\texttt{Online\_train\_tree}$'s lines 16 and 18, if node u gets updated, the $\texttt{left}$ (resp. $\texttt{right}$) entry of the binary cost vector $c^{\mathtt{u}}(\texttt{left})$ (resp. $c^{\mathtt{u}}(\texttt{right})$) takes value as u.left.cost (resp. u.right.cost), which is $\tilde{c}(\mathcal{T}^{\mathtt{u.left}}(x))$ (resp. $\tilde{c}(\mathcal{T}^{\mathtt{u.right}}(x))$). Therefore the binary CSMC example u receives is indeed $(x, c^{\mathtt{u}})$. This completes the proof of the lemma. $\square$

**Claim 18.** *For every level $d \in [D]$, denote by $\alpha_d$ and $\beta_d$ the ancestor of $\alpha, \beta$ at level $d$ in $\mathcal{T}$ respectively. Then, for node* v *at level $d$:*

1. If $\mathtt{v.id} < \alpha_d.\mathtt{id}$ *or* $\mathtt{v.id} > \beta_d.\mathtt{id}$, *then for all* $a \in \mathrm{range}(\mathcal{T}^{\mathtt{v}})$, $\tilde{c}(a) = 0$.

2. If $\alpha_d.\mathtt{id} < \mathtt{v.id} < \beta_d.\mathtt{id}$, *then for all* $a \in \mathrm{range}(\mathcal{T}^{\mathtt{v}})$, $\tilde{c}(a) = c^*$.

3. If $\mathtt{v.cost}$ *is available, it must equal* $\tilde{c}(\mathcal{T}^{\mathtt{v}}(x))$*; in addition,* $\mathtt{Return\_cost}(\mathtt{v}, \alpha_d, \beta_d)$ *returns* $\tilde{c}(\mathcal{T}^{\mathtt{v}}(x))$ *correctly.*

*Proof.* It can be seen that for every node $\mathtt{u}$ at level $d$, $\mathrm{range}(\mathcal{T}^{\mathtt{u}})$ spans a separate contiguous subinterval of $[0, 1]$. Specifically, for every $\mathtt{u}$ at level $d$, define interval

$$I_{\mathtt{u}} = \left[ \frac{\mathtt{u.id} - (2^d - 1)}{2^d}, \frac{\mathtt{u.id} + 1 - (2^d - 1)}{2^d} \right),$$

we have $\mathrm{range}(\mathcal{T}^{\mathtt{u}}) = \mathrm{range}(\mathcal{T}) \cap I_{\mathtt{u}}$, and all $I_{\mathtt{u}}$'s are disjoint for $\mathtt{u}$'s at level $d$.

For the first item, suppose $\mathtt{v.id} < \alpha_d.\mathtt{id}$, i.e. $\mathtt{v}$ is to the left of $\alpha_d$. In this case, all elements of $\mathrm{range}(\mathcal{T}^{\mathtt{v}})$ must be less than $a_{\min}$, and therefore for all $a \in \mathrm{range}(\mathcal{T}^{\mathtt{v}})$, $c^{\mathtt{v}}(a) = 0$. A similar reasoning applies to the case when $\mathtt{v.id} > \beta_d.\mathtt{id}$.

For the second item, suppose $\alpha_d.\mathtt{id} < \mathtt{v.id} < \beta_d.\mathtt{id}$, i.e. $\mathtt{v}$ is in the middle of $\alpha_d$ and $\beta_d$. In this case, all elements of $\mathrm{range}(\mathcal{T}^{\mathtt{v}})$ must be within the interval $[a_{\min}, a_{\max}]$, therefore, by the definition of $a_{\min}$ and $a_{\max}$, we have that for all $a \in \mathrm{range}(\mathcal{T}^{\mathtt{v}})$, $c^{\mathtt{v}}(a) = c^*$.

For the last item, we consider two cases.

1. If $\mathtt{v} \neq \alpha_d$ and $\mathtt{v} \neq \beta_d$, then from the first two items we have just shown, we can decide the value of $c^{\mathtt{v}}(\mathcal{T}^{\mathtt{v}}(x))$ directly by comparison with the $\mathtt{id}$'s of $\alpha$ and $\beta$, which is consistent with the implementation of $\mathtt{Return\_cost}$; also note that in this case, $\mathtt{v.cost}$ gets assigned to $\mathtt{Return\_cost}(\mathtt{v}, \alpha_d, \beta_d)$, which also equals $c^{\mathtt{v}}(\mathcal{T}^{\mathtt{v}}(x))$.

2. Otherwise, $\mathtt{v} = \alpha_d$ or $\mathtt{v} = \beta_d$. In this case, $\mathtt{Return\_cost}$ returns the stored cost of $\mathtt{v}$, i.e. $\mathtt{v.cost}$. It suffices to show that $\alpha_d.\mathtt{cost}$ (resp. $\beta_d.\mathtt{cost}$), is indeed $\tilde{c}(\mathcal{T}^{\alpha_d}(x))$ (resp. $\tilde{c}(\mathcal{T}^{\beta_d}(x))$), which we show by induction:

   **Base case.** In the case when $d = D$, $\alpha_D.\mathtt{cost} = \alpha.\mathtt{cost}$ (resp. $\beta_D.\mathtt{cost} = \beta.\mathtt{cost}$) is directly calculated in line 2 of Algorithm 10, and is indeed $\tilde{c}(\mathrm{label}(\alpha)) = c^{\mathtt{v}}(\alpha)$ (resp. $\tilde{c}(\mathrm{label}(\beta)) = c^{\mathtt{v}}(\beta)$), and is equal to $c^*$.

   **Inductive case.** Suppose for level $d + 1$, $\mathtt{Return\_cost}(\mathtt{u}, \alpha_{d+1}, \beta_{d+1})$ returns $c(\mathcal{T}^{\mathtt{u}}(x))$ correctly for $\mathtt{u}$ in $\{\alpha_{d+1}, \beta_{d+1}\}$. Now consider a node $\mathtt{v}$ at level $d$, which is either $\alpha_d$ or $\beta_d$. By inductive hypothesis, and the correctness of $\mathtt{Return\_cost}$ on the costs of non-ancestors of $\alpha, \beta$ in the last item, for both $\mathtt{v.left}$ and $\mathtt{v.right}$, their costs $c^{\mathtt{v}}(\mathtt{left}) = \tilde{c}(\mathcal{T}^{\mathtt{v.left}}(x))$ and $c^{\mathtt{v}}(\mathtt{right}) = \tilde{c}(\mathcal{T}^{\mathtt{v.right}}(x))$ are calculated correctly by $\mathtt{Return\_cost}$. Hence, the cost calculated by $\mathtt{Return\_cost}$ on node $\mathtt{v}$, $\mathtt{v.cost}$, at line 20 in $\mathtt{Online\_train\_tree}$, equals $c^{\mathtt{v}}(f^{\mathtt{v}}(x)) = \tilde{c}(\mathcal{T}^{\mathtt{v}.f^{\mathtt{v}}(x)}(x)) = \tilde{c}(\mathcal{T}^{\mathtt{v}}(x))$. This completes the induction.

The proof of the last item is complete. $\qquad\qquad\qquad\qquad\qquad\qquad\qquad\qquad\qquad$ $\square$

### G.3 Proof of Theorem 4

We are now ready to prove the time complexity guarantee of CATS, i.e. Theorem 4 in the main body.

*Proof of Theorem 4.* From Lemma 17, we see that $\mathtt{Online\_train\_tree}$ faithfully implements $\mathtt{Train\_tree}$ in an online fashion. As other steps of CATS are intact, the online implementation of CATS faithfully implements the original CATS.

Moreover, consider the operations of CATS at every time step:

1. Predict $\mathcal{T}(x)$: this takes $\mathcal{O}(D) = \mathcal{O}(\log K)$ time as can be directly seen from Algorithm 4.

2. Generate $\epsilon$-greedy action distribution, take action, create $(x_t, \tilde{c}_t)$ implicitly by representing $\tilde{c}_t$ as $(a_{\min}, a_{\max}, c^*)$: these steps take $\mathcal{O}(1)$ time as they are based on manipulations of piecewise constant density with at most 3 pieces.

3. `Online_train_tree`$(\mathcal{T}, (x_t, \tilde{c}_t))$: this takes $\mathcal{O}(D) = \mathcal{O}(\log K)$ time, because at each of the $D$ levels, there are at most 2 nodes to be updated, and for every such node, `Return_cost` takes $\mathcal{O}(1)$ time to retrieve the costs of both subtrees.

In summary, the total time cost of CATS at every time step is $\mathcal{O}(\log K)$. $\qquad\square$

## H   Additional Experimental Results

Additional figures comparing running times of CATS against `dLinear` and `dTree` for the rest of the datasets are shown in Figures 3-7.

Figure 3: Training time of CATS (blue bar) w.r.t: (**left**) bandwidth ($h$) with a fixed discretization scale $K = 2^{13}$; (**middle**) discretization scale ($1/K$) with a fixed $h = 1/4$; (**right**) discretization scale ($1/K$) with a fixed $h = 1/4$, compared against `dLinear` (orange bar) and `dTree` (green bar), in the `cpu_act` dataset.

Figure 4: Training time of CATS (blue bar) w.r.t: (**left**) bandwidth ($h$) with a fixed discretization scale $K = 2^{13}$; (**middle**) discretization scale ($1/K$) with a fixed $h = 1/4$; (**right**) discretization scale ($1/K$) with a fixed $h = 1/4$, compared against `dLinear` (orange bar) and `dTree` (green bar), in the `zurich_delay` dataset.

Figure 5: Training time of CATS (blue bar) w.r.t: (**left**) bandwidth ($h$) with a fixed discretization scale $K = 2^{13}$; (**middle**) discretization scale ($1/K$) with a fixed $h = 1/4$; (**right**) discretization scale ($1/K$) with a fixed $h = 1/4$, compared against `dLinear` (orange bar) and `dTree` (green bar), in the `wisconsin` dataset.

Figure 6: Training time of CATS (blue bar) w.r.t: (**left**) bandwidth ($h$) with a fixed discretization scale $K = 2^{13}$; (**middle**) discretization scale ($1/K$) with a fixed $h = 1/4$; (**right**) discretization scale ($1/K$) with a fixed $h = 1/4$, compared against dLinear (orange bar) and dTree (green bar), in the black_friday dataset.

Figure 7: Training time of CATS (blue bar) w.r.t: (**left**) bandwidth ($h$) with a fixed discretization scale $K = 2^{13}$; (**middle**) discretization scale ($1/K$) with a fixed $h = 1/4$; (**right**) discretization scale ($1/K$) with a fixed $h = 1/4$, compared against dLinear (orange bar) and dTree (green bar), in the auto_price dataset.

## Footnotes

[10]subject to $T \to \infty$.