[Reviews · NeurIPS 2020]

Review 1

Summary and Contributions: This paper present an extension of Contextual Bandits to continuous action spaces. Them main idea is to simultaneously discretize the action space into a regular 1-D grid and to smoothen the policy, by allowing an action to be drawn randomly from an interval centered around the grid points. One key point is to ensure the efficiency of the algorithm: this is realised by reducing the problem to building a binary tree of cost-sensitive multi-class classifiers, that can be trained efficiently and incrementally on-line. A computational complexity of O(logK) is achieved, where K is the discretization level.

Strengths: * This work has strong theoretical grounding and, to the best of my knowledge, is quite novel in the way it addresses both the efficiency and discretization issues. * The approach (by reduction to a set of cost-sensitive multi-class classification problems) is elegant and simple.

Weaknesses: * Relevant to a minority of the NeurIPS community * The experimental section -limited to on-line regression problems - is a bit disappointing. The experimental part should have addressed true and realistic contextual bandits problems (e.g. all the concrete examples and cases cited in the second paragraph of the introduction; reinforcement learning problems). * There are some important hyper-parameters in the method. How to choose them in function of the task and the dataset (except by brute-force, as it is done in Algorithm 3) remains an open issue. EDIT AFTER Rebuttal: @authors: it would be great if you can include the paragraph in the rebuttal that handles that point in the paper as well.

Correctness: Claims, theorems and proofs seem to be correct (to the best of my understanding and knowledge).

Clarity: The paper is well written: motivations are clearly exposed, ideas and methods are well structured and the paper is, globally, easy to read (even if some sections need to be read two times before being understandable).

Relation to Prior Work: To the best of my knowledge, the paper explains clearly the landscape of the current state-of-the-art and how the proposed approach differs from it.

Reproducibility: Yes

Additional Feedback: It will be useful to better describe the link and relationship between the choice of h (smoothing bandwidth) and K (discretization level), and how this choice influences the bias/variance trade-off. Minor comments: * Line 110: add 'h' subscript to 'Smooth' (Smooth --> Smooth_h) * Line 194: replace 'line 5 in CATS' by 'line 5 in Train_Tree) * Line 6 of Algo3: replace $T_{t,h}^{...}$ by $T_{t}^{...}$. BTW, why to draw randomly from the set and not to choose the last one (i.e t=T)?


Review 2

Summary and Contributions: ---- I have read the rebuttal and I am still convinced that it is a good paper and a clear accept. The paper reduces contextual bandits with continuous actions places to cost sensitive classification over a discretized set of actions followed by uniform smoothing. The algorithm is made efficient by making the policy class to be a hierarchical tree policy class, where at each internal mode there is a binary classification pool-icy from some class F. The algorithm has a train and prediction cost of O(log K) per example (K is the number of discretized arms). Regret guarantees are provided in the realizable setting. Some empirical comparisons are done comparing the algorithm to contextual bandits with policy class dTree and standard one vs all.

Strengths: 1. The algorithm is a reduction style algorithm. Thus any progress ins tree based extreme classification can be carried over to this approach in a seem-less manner. 2. The proposed algorithm has a computational training cost and inference cost of O(log K) per sample. This can support a much finer discretization of the space in a computationally efficient manner. 3. Formal regret guarantees are provided for the algorithm. Except for the K^{2/3} dependence, the guarantee seems to be optimal.

Weaknesses: 1. The paper is very terse in some important areas which made it difficult to understand some of the claims. For instance, I do not fully understand the point being made in line 208 to 213. It would be great of the authors add an example as to why the cost of finding the nodes which need to be updated is only O(log K). 2. The dependency on K in the regret if O(K^{2/3}), which does not seem to be optimal as mentioned in the paper as well. Intuitively epoch-greedy algorithm should have the usual K^{1/3} dependency. However, I do not fully understand the explanation or guess provided in the paper wrt why this happens. In the realizable setting, what exactly is the approximation in Train_tree in terms of computing the ERM? It would be great if the authors can add some explanation regarding this. 3. It is not clear what exactly the dTree based algorithm is in the empirical results. Is the algorithm just discretizing the set and treating them as K arms. Then the filter-tree algorithm is used as the function class for contextual bandits?

Correctness: I think the claims are correct. The empirical methodology is correct and adequate for the setting. It would however be a bonus to compare with CGP-UCB style algorithms.

Clarity: The paper is clearly written modulo the terse-ness related issues I mentioned above.

Relation to Prior Work: Prior work is discussed adequately.

Reproducibility: Yes

Additional Feedback:


Review 3

Summary and Contributions: +++++ The authors have done a good job on the rebuttal, I assume they will update correspondingly according to all comments and suggestions, hence I increased my score. +++++ This submission tries to study stale contextual bandits with continuous actions having the unknown structure to make you choose the best action given the context and comparing it with dLinear and dTree.

Strengths: 1. this draft is easy to understand and follow that extends from [35]'s data-dependent regret bounds as[2] 2. your core idea of smoothing of /epsilon-greedy as[3] and discrete action space is expected 3. training tree is a subset of graphical structure like clusters of graphs found

Weaknesses: 1. your abstract need to rewrite to describe what are the main challenges and how you solve them in order to advance this field 2. your contributions are problematic, you may want to write, why you propose those are way more important 3. in fig 1 your proposed method achieves performance similar to dLinear

Correctness: Yes

Clarity: Readable, but away from high quality

Relation to Prior Work: Not yet

Reproducibility: Yes

Additional Feedback: 1. baselines are too few and weak, related state-of-the-art that replaced traditional contextual bandits you may want to compare to demonstrate your significance: [1]Fast Distributed Bandits for Online Recommendation Systems, [2]On Context-Dependent Clustering of Bandits, [3]Improved Algorithm on Online Clustering of Bandits, especially with [2,3]'s regret analysis that sharing the same spirit of data-dependent and exploration 2. about naive implementation and corresponding places in the draft you may remove that is trivial to save some spaces 3. your main assumptions are impractical and unrealistic, Theorems 6 and 7 their practical insights are unclear 4. your experimental results are not significant, e.g., data sets' brief description should be described, and you relay on a specific platform which is another drawback, you may want to report generic experimental environments for a large-scale setting as[1] to show your practicalness 5. your conclusion is not well-shaped too, overall writing throughout can be largely improved


Review 4

Summary and Contributions: The paper proposes a computationally tractable algorithm for contextual bandits with continuous actions. The algorithm is a tree algorithm that has sub-linear regret with respect to the tree policy class under realizability assumption. An off-policy version of the algorithm is also proposed and analyzed. Simulation experiments demonstrate that the algorithm has superior performance compared to other benchmark algorithms.

Strengths: The problem studied is very relevant to the NeurIPS community. The results showed in this paper seem significant because of the exponential improvement in the computational complexity of the proposed algorithm. The argument is convincing because the simulation experiments verify that the proposed algorithm performs similarly to an algorithm with exponentially worse complexity argument, and performs better than an algorithm with similar complexity. It is also good to see that off-policy optimization works for the proposed algorithm because theoretically and empirically.

Weaknesses: I cannot think of any obvious limitation.

Correctness: I am not familiar with relevant literature on tree policies or discretized algorithms for bandits so I am unable to verify the details of the proofs. The theoretical claims look correct and the simulation experiments are convincing.

Clarity: Perhaps because I am not familiar with relevant literature, I find the discussion section starting on line 47 a little bit hard to follow. Is the “smoothing” approach the main innovation? When will the best single action impossible to find? Why do we need to guess width but the location can be automatically adjusted? Other than this, the paper is in general well-written.

Relation to Prior Work: Yes, the relation to prior work is clearly discussed.

Reproducibility: Yes

Additional Feedback: I have a few questions for the authors. 1. Is it possible to theoretically show or at least empirically check that the thee realizability assumption is satisfied? 2. Why are the confidence intervals calculated with a single run instead of multiple runs? Is it because of the computational time or resources? 3. Line 264 states that you believe the suboptimal dependence on K is a price to pay for using the computationally efficient algorithm, is there any evidence to support this claim? ------------------ The author's response answers my questions. But I would like to keep my score the same.

[Author Response · NeurIPS 2020]

We thank all reviewers for their thoughtful comments.

**Experimental setup (R1, R3):** The reviewers brought up an excellent suggestion to deploy our algorithm in production
on real users, for example using a user-interactive system. We agree this is an appropriate next step for development.
However, applying contextual bandits to continuous-action domains is a relatively new approach; the large-scale
contextual bandit datasets for the examples in the introduction have not been collected yet.

Simulating contextual bandit learning using regression/classification datasets is a standard protocol for academic
evaluation of bandit algorithms [see e.g. A. Bietti et al. A Contextual Bandit Bake-off]. Note that we have chosen
the regression datasets with the criterion of having million scale examples with unique regression values, so it is very
relevant to a real contextual bandit setting.

**Approximate ERM guarantee of `Train_tree` (R2, R4):** At a high level, our analysis of the `Train_tree` algorithm
needs to account for compounding errors accumulated in each of the nodes, which results in a $O(K\sqrt{1/n})$ rate slower
than the $O(\sqrt{K/n})$ rate achieved by ERM. We will clarify this in the final version.

**Realizability assumption (R3, R4):** Indeed we assume realizability in Theorems 6 and 7. This is a typical situation in
ML theory when the theory does not necessarily capture *all* cases when things work, but instead gives justification and
assurance that they do work in practice.

Realizability has been widely used in many successful contextual bandit algorithms (e.g. LinUCB, OFUL).

Regarding checking realizability: This argument applies to essentially all supervised learning settings, where standard
surrogate loss minimization can fail without realizability or related unverifiable assumptions.

We now answer specific questions raised by individual reviewers:

**R1:** We agree that in our online contextual bandit learning setting, tuning hyperparameters $h$ and $K$ can be problematic.
Note that all other hyperparameters such as learning rate, the greedy parameter, and the penalty term in line 5
of Algorithm 3 (`CATS_Off`) are fixed in our experiments and for all of the datasets. For practical applications, we
recommend using `CATS_Off` to perform off-policy optimization to find the best $(h, K)$ combination, whose effectiveness
is demonstrated in Figure 1 (right) in the paper. Note that in `CATS_Off`, we only use input logged data to select the best
$h, K$ hyperparameters.

Regarding the choice of $h$ and $K$ in bias-variance trade-off: Tuning $K$ has the usual bias-complexity trade-off in
standard statistical learning. If we are competing with the best (non-smoothed) policy in a class, then tuning $h$ also
incurs a bias-variance tradeoff: (1) the smoothed loss more closely approximates the true loss for smaller $h$; (2) a
smaller $h$ makes achieving a low $h$-smoothed regret bound harder.

**R2:** The reviewer's understanding regarding the `dTree` baseline and the filter tree algorithm is correct. We will clarify
this in the final version. We will elaborate more on line 208-213 in the final version.

**R3:** The reviewer mentioned our accuracy is similar to the `dLinear` baseline. This is true, but our algorithm beats
`dLinear` in terms of time complexity. As R4 has acknowledged, our algorithm wins over `dLinear` computationally
and wins over `dTree` statistically.

The reviewer has suggested three papers to compare our algorithm with. Note that [1] was published *after* the submission
deadline so we could not have possibly compared to it. [2] and [3] are also not directly relevant to our work since (1)
they impose structure on contexts, rather than actions, and moreover the structure is *linear* rather than Lipschitz; (2)
they work with discrete action spaces. We thank the reviewer for bringing these papers to our attention and will cite
them in our final draft.

We will polish the writing on the sections that the reviewer highlighted in the final version.

**R4:** The idea of smoothing was initially proposed in [35], but their proposed algorithm is computationally intractable.
Our contribution is proposing a computationally efficient algorithm for continuous action spaces in the smoothing
framework. Our online algorithm `CATS`, its off-policy optimization version `CATS_Off`, and their statistical guarantees
as well as our careful implementation design with $O(\log K)$ are all new and constitute the contributions of this work.

See Figure 1 of [35] for an example of a discontinuous loss function, where the best single action is impossible to find
within a finite number of rounds of contextual bandit learning.

Regarding the reason we need to guess width but the location can be automatically adjusted: If the learner sets a high
discretization level $K$ then it can take actions in [0,1] with fine granularity.

[Meta-Review · NeurIPS 2020]

The paper makes significant contributions, as brought up by the reviewers, towards providing efficient approaches for large (continuous-action) contextual bandits. The author response has also aided the understanding of the paper and provided several clarifications. The author(s) is/are urged to incorporate the reviewers' suggestions appropriately in preparing the final version.